# Designing Robust Transformers using Robust Kernel Density Estimation

**Xing Han**
Department of ECE
University of Texas at Austin
aaronhan223@utexas.edu

**Tongzheng Ren**
Department of Computer Science
University of Texas at Austin
tongzheng@utexas.edu

**Tan Minh Nguyen**
Department of Mathematics
University of California, Los Angeles
tanmnguyen89@ucla.edu

**Khai Nguyen**
Department of Statistics and Data Sciences
University of Texas at Austin
khainb@utexas.edu

**Joydeep Ghosh**
Department of ECE
University of Texas at Austin
jghosh@utexas.edu

**Nhat Ho**
Department of Statistics and Data Sciences
University of Texas at Austin
minhnhat@utexas.edu

## Abstract

Transformer-based architectures have recently exhibited remarkable successes across different domains beyond just powering large language models. However, existing approaches typically focus on predictive accuracy and computational cost, largely ignoring certain other practical issues such as robustness to contaminated samples. In this paper, by re-interpreting the self-attention mechanism as a non-parametric kernel density estimator, we adapt classical robust kernel density estimation methods to develop novel classes of transformers that are resistant to adversarial attacks and data contamination. We first propose methods that down-weight outliers in RKHS when computing the self-attention operations. We empirically show that these methods produce improved performance over existing state-of-the-art methods, particularly on image data under adversarial attacks. Then we leverage the median-of-means principle to obtain another efficient approach that results in noticeably enhanced performance and robustness on language modeling and time series classification tasks. Our methods can be combined with existing transformers to augment their robust properties, thus promising to impact a wide variety of applications.

## 1 Introduction

Attention mechanisms and transformers (Vaswani et al., 2017a) have drawn lots of attention in the machine learning community (Lin et al., 2021; Tay et al., 2020; Khan et al., 2021). Now they are among the best deep learning architectures for a variety of applications, including those in natural language processing (Devlin et al., 2019; Al-Rfou et al., 2019; Dai et al., 2019; Child et al., 2019; Raffel et al., 2020; Baevski & Auli, 2019; Brown et al., 2020; Dehghani et al., 2019), computer vision (Dosovitskiy et al., 2021; Liu et al., 2021b; Touvron et al., 2021a; Ramesh et al., 2021; Radford et al., 2021; Fan et al., 2021; Liu et al., 2022), and reinforcement learning (Chen et al., 2021; Janner et al., 2021). They are also known for their effectiveness in transferring knowledge from various pretraining tasks to different downstream applications with weak supervision or no supervision (Radford et al., 2018, 2019; Devlin et al., 2019; Yang et al., 2019; Liu et al., 2019).

37th Conference on Neural Information Processing Systems (NeurIPS 2023).

While there have been notable advancements, the robustness of the standard attention module remains an unresolved issue in the literature. In this paper, our goal is to reinforce the attention mechanism and construct a comprehensive framework for robust transformer models. To achieve this, we first revisit the interpretation of self-attention in transformers, viewing it through the prism of the Nadaraya-Watson (NW) estimator (Nadaraya, 1964) in a non-parametric regression context. Within the transformer paradigm, the NW estimator is constructed based on the kernel density estimators (KDE) of the keys and queries. However, these KDEs are not immune to the issue of sample contamination (Kim & Scott, 2012). By conceptualizing the KDE as a solution to the kernel regression problem within a Reproducing Kernel Hilbert Space (RKHS), we can utilize a range of state-of-the-art robust KDE techniques, such as those based on robust kernel regression and median-of-mean estimators. This facilitates the creation of substantially more robust self-attention mechanisms. The resulting suite of robust self-attention can be adapted to a variety of transformer architectures and tasks across different data modalities. We carry out exhaustive experiments covering vision, language modeling, and time-series classification. The results demonstrate that our approaches can uphold comparable accuracy on clean data while exhibiting improved performance on contaminated data. Crucially, this is accomplished without introducing any extra parameters.

**Related Work on Robust Transformers:** Vision Transformer (ViT) models (Dosovitskiy et al., 2020; Touvron et al., 2021b) have recently demonstrated impressive performance across various vision tasks, positioning themselves as a compelling alternative to CNNs. A number of studies (e.g., Subramanya et al., 2022; Paul & Chen, 2022; Bhojanapalli et al., 2021; Mahmood et al., 2021; Mao et al., 2022; Zhou et al., 2022) have proposed strategies to bolster the resilience of these models against common adversarial attacks on image data, thereby enhancing their generalizability across diverse datasets. For instance, Mahmood et al. (2021) provided empirical evidence of ViT's vulnerability to white-box adversarial attacks, while demonstrating that a straightforward ensemble defense could achieve remarkable robustness without compromising accuracy on clean data. Zhou et al. (2022) suggested fully attentional networks to enhance self-attention, achieving state-of-the-art accuracy on corrupted images. Furthermore, Mao et al. (2022) conducted a robustness analysis on various ViT building blocks, proposing position-aware attention scaling and patch-wise augmentation to enhance the model's robustness and accuracy. However, these investigations are primarily geared toward vision-related tasks, which restricts their applicability across different data modalities. As an example, the position-based attention from Mao et al. (2022) induces a bi-directional information flow, which is limiting for position-sensitive datasets such as text or sequences. These methods also introduce additional parameters. Beyond these vision-focused studies, robust transformers have also been explored in fields like text analysis and social media. Yang et al. (2022) delved into table understanding and suggested a robust, structurally aware table-text encoding architecture to mitigate the effects of row and column order perturbations. Liu et al. (2021a) proposed a robust end-to-end transformer-based model for crisis detection and recognition. Furthermore, Li et al. (2020) developed a unique attention mechanism to create a robust neural text-to-speech model capable of synthesizing both natural and stable audios. We have noted that these methods vary in their methodologies, largely due to differences in application domains, and therefore limiting their generalizability across diverse contexts.

**Other Theoretical Frameworks for Attention Mechanisms:** Attention mechanisms in transformers have been recently studied from different perspectives. Tsai et al. (2019) show that attention can be derived from smoothing the inputs with appropriate kernels. Katharopoulos et al. (2020); Choromanski et al. (2021); Wang et al. (2020) further linearize the softmax kernel in attention to attain a family of efficient transformers with both linear computational and memory complexity. These linear attentions are proven in Cao (2021) to be equivalent to a Petrov-Galerkin projection (Reddy, 2004), thereby indicating that the softmax normalization in dot-product attention is sufficient but not necessary. Other frameworks for analyzing transformers that use ordinary/partial differential equations include Lu et al. (2019); Sander et al. (2022). In addition, the Gaussian mixture model and graph-structured learning have been utilized to study attentions and transformers (Tang & Matteson, 2021; Gabbur et al., 2021; Zhang & Feng, 2021; Wang et al., 2018; Kreuzer et al., 2021). Nguyen et al. (2022c) has linked the self-attention mechanism with a non-parametric regression perspective, which offers enhanced interpretability of Transformers. Our approach draws upon this viewpoint, but focuses instead on how it can lead to robust solutions.

## 2 Self-Attention Mechanism from a Non-parametric Regression Perspective

Assume we have the key and value vectors $\{\boldsymbol{k}_j, \mathbf{v}_j\}_{j \in [N]}$ that is collected from the data generating process $\mathbf{v} = f(\boldsymbol{k}) + \varepsilon$, where $\varepsilon$ is some noise vectors with $\mathbb{E}[\varepsilon] = 0$, and $f$ is the function that we want to estimate. We consider a random design setting where the key vectors $\{\boldsymbol{k}_j\}_{j \in [N]}$ are i.i.d. samples from the distribution $p(\boldsymbol{k})$, and we use $p(\mathbf{v}, \boldsymbol{k})$ to denote the joint distribution of $(\mathbf{v}, \boldsymbol{k})$ defined by the data generating process. Our target is to estimate $f(\boldsymbol{q})$ for any new queries $\boldsymbol{q}$. The NW estimator provides a non-parametric approach to estimating the function $f$, the main idea is that

$$f(\boldsymbol{k}) = \mathbb{E}[\mathbf{v}|\boldsymbol{k}] = \int_{\mathbb{R}^D} \mathbf{v} \cdot p(\mathbf{v}|\boldsymbol{k}) d\mathbf{v} = \int_{\mathbb{R}^D} \frac{\mathbf{v} \cdot p(\mathbf{v}, \boldsymbol{k})}{p(\boldsymbol{k})} d\mathbf{v}, \tag{1}$$

where the first equation comes from the fact that $\mathbb{E}[\varepsilon] = 0$, the second equation comes from the definition of conditional expectation, and the last equation comes from the definition of conditional density. To provide an estimation of $f$, we just need to obtain estimations for both the joint density function $p(\mathbf{v}, \boldsymbol{k})$ and the marginal density function $p(\boldsymbol{k})$. KDE is commonly used for the density estimation problem (Rosenblatt, 1956; Parzen, 1962), which requires a kernel $k_\sigma$ with the bandwidth parameter $\sigma$ satisfies $\int_{\mathbb{R}^D} k_\sigma(\boldsymbol{x} - \boldsymbol{x}') d\boldsymbol{x} = 1, \forall \boldsymbol{x}'$, and estimate the density as

$$\hat{p}_\sigma(\mathbf{v}, \boldsymbol{k}) = \frac{1}{N} \sum_{j \in [N]} k_\sigma([\mathbf{v}, \boldsymbol{k}] - [\mathbf{v}_j, \boldsymbol{k}_j]) \quad \hat{p}_\sigma(\boldsymbol{k}) = \frac{1}{N} \sum_{j \in [N]} k_\sigma(\boldsymbol{k} - \boldsymbol{k}_j), \tag{2}$$

where $[\mathbf{v}, \boldsymbol{k}]$ denotes the concatenation of $\mathbf{v}$ and $\boldsymbol{k}$. Specifically, when $k_\sigma$ is the isotropic Gaussian kernel, we have $\hat{p}_\sigma(\mathbf{v}, \boldsymbol{k}) = \frac{1}{N} \sum_{j \in [N]} k_\sigma(\mathbf{v} - \mathbf{v}_j) k_\sigma(\boldsymbol{k} - \boldsymbol{k}_j)$. Combine this with Eq. (1) and Eq. (2), we can obtain the NW estimator of the function $f$ as

$$\widehat{f}_\sigma(\boldsymbol{k}) = \frac{\sum_{j \in [N]} \mathbf{v}_j k_\sigma(\boldsymbol{k} - \boldsymbol{k}_j)}{\sum_{j \in [N]} k_\sigma(\boldsymbol{k} - \boldsymbol{k}_j)}. \tag{3}$$

Furthermore, it is not hard to show that if the keys $\{\boldsymbol{k}_j\}_{j \in [N]}$ are normalized, the self-attention mechanism $\widehat{f}_\sigma(\boldsymbol{q}_i)$ in Eq. (3) is exactly the standard Softmax attention $\widehat{f}_\sigma(\boldsymbol{q}_i) = \sum_{j \in [N]} \text{softmax}(\boldsymbol{q}^\top \boldsymbol{k}_j / \sigma^2) \mathbf{v}_j$. Such an assumption on the normalized key $\{\boldsymbol{k}_j\}_{j \in [N]}$ can be mild, as in practice we always have a normalization step on the key to stabilizing the training of the transformer (Schlag et al., 2021). If we choose $\sigma^2 = \sqrt{D}$, where $D$ is the dimension of $\boldsymbol{q}$ and $\boldsymbol{k}_j$, then $\widehat{f}_\sigma(\boldsymbol{q}_i) = \boldsymbol{h}_i$. As a result, the self-attention mechanism in fact performs a non-parametric regression with NW-estimator and isotropic Gaussian kernel when the keys are normalized.

**KDE as a Regression Problem in RKHS** We start with the formal definition of the RKHS. The space $\mathcal{H}_k = \{f \mid f : \mathcal{X} \to \mathbb{R}\}$ is the RKHS associated with kernel $k$, where $k : \mathcal{X} \times \mathcal{X} \to \mathbb{R}$, if it is a Hilbert space with inner product $\langle \cdot, \cdot \rangle_{\mathcal{H}_k}$ and following properties:

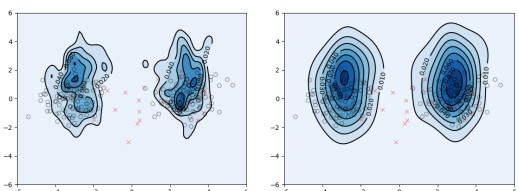

- $k(\boldsymbol{x}, \cdot) \in \mathcal{H}_k, \forall \boldsymbol{x} \in \mathcal{X}$;
- $\forall f \in \mathcal{H}_k, f(\boldsymbol{x}) = \langle f, k(\boldsymbol{x}, \cdot) \rangle_{\mathcal{H}_k}$. Aka the reproducing property.

With slightly abuse of notation, we define $k_\sigma(\boldsymbol{x}, \boldsymbol{x}') = k_\sigma(\boldsymbol{x} - \boldsymbol{x}')$. By the definition of the RKHS and the KDE estimator, we know $\hat{p}_\sigma = \frac{1}{N} \sum_{j \in [N]} k_\sigma(\boldsymbol{x}_j, \cdot) \in \mathcal{H}_{k_\sigma}$, and can be viewed as the optimal solution of the following least-square regression problem in RKHS:

Figure 1: The contour plots illustrate the density estimation of the two-dimensional query vector embedding within a transformer's attention layer. The left plot employs the regular KDE method, as defined in Eq. (4), whereas the right plot utilizes a robustified version of the KDE method, which enhances KDE's robustness against outliers.

$$\hat{p}_\sigma = \arg\min_{p \in \mathcal{H}_{k_\sigma}} \sum_{j \in [N]} \frac{1}{N} \|k_\sigma(\boldsymbol{x}_j, \cdot) - p\|^2_{\mathcal{H}_{k_\sigma}}. \tag{4}$$

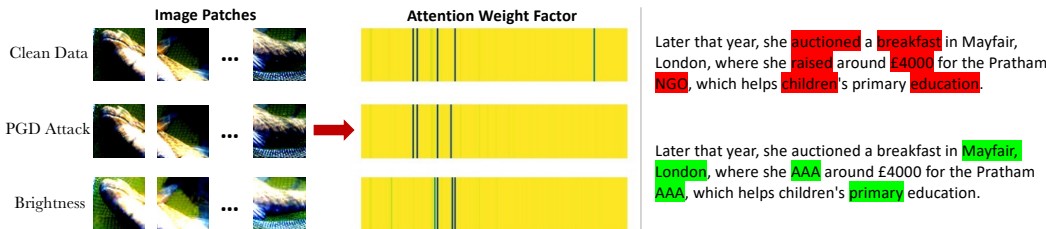

Figure 2: The application of Transformers with robust KDE attention on image and text is shown. (Left) The robust KDE self-attention generates varying weight factors for image patch embeddings under adversarial attacks or data corruption. The adversely impacted regions that would otherwise lead to incorrect predictions are down-weighted, ensuring enhanced accuracy and robustness. (Right) In the field of language modeling, the weight factors lend significance to essential keywords (highlighted in red). In the face of word swap attacks, the fortified self-attention mechanism, particularly when utilizing the medians-of-means principle, is proficient in disregarding or reducing the importance of less consequential words (marked in green). Consequently, this results in a more resilient procedure during self-attention computations.

Note that, in Eq. (4), the same weight factor $1/N$ is applied uniformly to each error term $\|k_\sigma(\boldsymbol{x}_j, \cdot) - p\|^2_{\mathcal{H}_{k_\sigma}}$. This approach functions effectively if there are no outliers in the set $\{k_\sigma(\boldsymbol{x}_j, \cdot)\}_{j \in [N]}$. However, when outliers are present (for instance, when there is some $j$ such that $\|k_\sigma(\boldsymbol{x}_j, \cdot)\|_{\mathcal{H}_{k_\sigma}} \gg \|k_\sigma(\boldsymbol{x}_i, \cdot)\|_{\mathcal{H}_{k_\sigma}}, \forall i \in [N], i \neq j$), the error attributable to these outliers will overwhelmingly influence the total error, leading to a significant deterioration in the overall density estimation. We illustrate the robustness issue of the KDE in Figure 1. The view that KDE is susceptible to outliers, coupled with the non-parametric understanding of the self-attention mechanism, implies a potential lack of robustness in Transformers when handling outlier-rich data. We now offer a fresh perspective on this robustness issue, introducing a universal framework that is applicable across diverse data modalities.

## 3 Robust Transformers that Employ Robust Kernel Density Estimators

Drawing on the non-parametric regression formulation of self-attention, we derive multiple robust variants of the NW-estimator and demonstrate their applicability in fortifying existing Transformers. We propose two distinct types of robust self-attention mechanisms and delve into the properties of each, potentially paving the way for Transformer variants that are substantially more robust.

### 3.1 Down-weighting Outliers in RKHS

Inspired by robust regression (Fox & Weisberg, 2002), a direct approach to achieving robust KDE involves down-weighting outliers in the RKHS. More specifically, we substitute the least-square loss in Eq. (4) with a robust loss function $\rho$, resulting in the following formulation:

$$\hat{p}_{\text{robust}} = \underset{p \in \mathcal{H}_{k_\sigma}}{\arg\min} \sum_{j \in [N]} \rho \left( \|k_\sigma(\boldsymbol{x}_j, \cdot) - p\|_{\mathcal{H}_{k_\sigma}} \right) = \sum_{j \in [N]} \omega_j k_\sigma(\boldsymbol{x}_j, \cdot). \tag{5}$$

Examples of the robust loss function $\rho$ include the Huber loss (Huber, 1992), Hampel loss (Hampel et al., 1986), Welsch loss (Welsch & Becker, 1975) and Tukey loss (Fox & Weisberg, 2002). We empirically evaluate different loss functions in our experiments. The critical step here is to estimate the set of weights $\omega = (\omega_1, \cdots, \omega_N) \in \Delta_N$, with each $\omega_j \propto \psi \left( \|k_\sigma(\boldsymbol{x}_j, \cdot) - \hat{p}_{\text{robust}}\|_{\mathcal{H}_{k_\sigma}} \right)$, where $\psi(x) := \frac{\rho'(x)}{x}$. Since $\hat{p}_{\text{robust}}$ is defined via $\omega$, and $\omega$ also depends on $\hat{p}_{\text{robust}}$, one can address this circular definition problem via an alternative updating algorithm proposed by Kim & Scott (2012). The algorithm starts with randomly initialized $\omega^{(0)} \in \Delta_n$, and performs alternative updates between $\hat{p}_{\text{robust}}$ and $\omega$ until the optimal $\hat{p}_{\text{robust}}$ is reached at the fixed point (see details in Appendix A).

However, while this technique effectively diminishes the influence of outliers, it also comes with noticeable drawbacks. Firstly, it necessitates the appropriate selection of the robust loss function, which may entail additional effort to understand the patterns of outliers. Secondly, the iterative

---

**Algorithm 1** Procedure of Computing Attention Vector of Transformer-RKDE/SPKDE/MoM

---

1: **Input**: $\mathbf{Q} = \{\boldsymbol{q}_i\}_{i\in[N]}$, $\mathbf{K} = \{\boldsymbol{k}_j\}_{j\in[N]}$, $\mathbf{V} = \{\mathbf{v}_l\}_{l\in[N]}$, initial weights $\omega^{(0)}$
2: Normalize $\mathbf{K} = \{\boldsymbol{k}_j\}_{j\in[N]}$ along the head dimension.
3: Compute kernel function between each pair of sequence: $k_\sigma(\mathbf{Q}, \mathbf{K}) = \{k_\sigma(\boldsymbol{q}_i - \boldsymbol{k}_j)\}_{i,j\in[N]}$.
4: (Optional) apply attention mask on $k_\sigma(\mathbf{Q}, \mathbf{K})$.
5: **[MoM]** Randomly sample $B$ subsets $I_1, \ldots, I_B$ of size $\mathcal{S}$, obtain the median block $I_l$ such that $\frac{1}{\mathcal{S}}\sum_{j\in I_l} k_\sigma(\boldsymbol{q}_i - \boldsymbol{k}_j) = \text{median}\{\frac{1}{\mathcal{S}}\sum_{j\in I_1} k_\sigma(\boldsymbol{q}_i - \boldsymbol{k}_j), \ldots, \frac{1}{\mathcal{S}}\sum_{j\in I_B} k_\sigma(\boldsymbol{q}_i - \boldsymbol{k}_j)\}$
6: **[RKDE]** Update weights $\omega^{(0)}$ for marginal/joint density by $\omega_j^{(1)} = \dfrac{\psi\left(\left\|k_\sigma(\boldsymbol{k}_j, \cdot) - \hat{p}_{\text{robust}}^{(k)}(\boldsymbol{k})\right\|_{\mathcal{H}_{k_\sigma}}\right)}{\sum_{j\in[N]} \psi\left(\left\|k_\sigma(\boldsymbol{k}_j, \cdot) - \hat{p}_{\text{robust}}^{(k)}(\boldsymbol{k})\right\|_{\mathcal{H}_{k_\sigma}}\right)}$.
7: **[SPKDE]** Obtain optimal weights $\omega$ for marginal/joint density via solving equation (7).
8: **[RKDE, SPKDE]** Obtain robust self-attention vector $\quad \widehat{\boldsymbol{h}}_i = \dfrac{\sum_{j\in[N]} \mathbf{v}_j \omega_j^{\text{joint}} k_\sigma(\boldsymbol{q}_i - \boldsymbol{k}_j)}{\sum_{j\in[N]} \omega_j^{\text{marginal}} k_\sigma(\boldsymbol{q}_i - \boldsymbol{k}_j)}$.
9: **[MoM]** Obtain attention vector $\widehat{\boldsymbol{h}}_i = \dfrac{\sum_{j\in I_l} v_j k_\sigma(\boldsymbol{q}_i - \boldsymbol{k}_j)}{\sum_{j\in I_l} k_\sigma(\boldsymbol{q}_i - \boldsymbol{k}_j)}$.

---

updates might not successfully converge to the optimal solution. A better alternative is to assign higher weights to high-density regions and reduce the weights for atypical samples. The original KDE is scaled and projected to its nearest weighted KDE according to the $L_2$ norm. Similar concepts have been studied by Scaled and Projected KDE (SPKDE) (Vandermeulen & Scott, 2014), which offer an improved set of weights that better defend against outliers in the RKHS space. Specifically, given the scaling factor $\beta > 1$, and let $\mathcal{C}_\sigma^N$ be the convex hull of $k_\sigma(\boldsymbol{x}_1, \cdot), \ldots, k_\sigma(\boldsymbol{x}_N, \cdot) \in \mathcal{H}_{k_\sigma}$, i.e., the space of weighted KDEs, the optimal density $\hat{p}_{\text{robust}}$ is given by

$$\hat{p}_{\text{robust}} = \arg\min_{p\in\mathcal{C}_\sigma^N} \left\| \frac{\beta}{N}\sum_{j\in[N]} k_\sigma(x_j, \cdot) - p \right\|_{\mathcal{H}_{k_\sigma}}^2, \tag{6}$$

which is guaranteed to have a unique minimizer since we are projecting in a Hilbert space and $\mathcal{C}_\sigma^N$ is closed and convex. Notice that, by definition, $\hat{p}_{\text{robust}}$ can also be represented as $\hat{p}_{\text{robust}} = \sum_{j\in[N]} \omega_j k_\sigma(x_j, \cdot)$, $\omega \in \Delta^N$, which is same as the formulation in Eq. (5). Then Eq. (6) can be written as a quadratic programming (QP) problem over $\omega$:

$$\min_\omega \omega^\top G\omega - 2q^\top\omega, \quad \text{subject to } \omega \in \Delta^N, \tag{7}$$

where $G$ is the Gram matrix of $\{\boldsymbol{x}_j\}_{j\in[N]}$ with $k_\sigma$ and $q = G\mathbf{1}\frac{\beta}{N}$. Since $k_\sigma$ is a positive-definite kernel and each $\boldsymbol{x}_i$ is unique, the Gram matrix $G$ is also positive-definite. As a result, this QP problem is convex, and we can leverage commonly used solvers to efficiently obtain the solution and the optimal density $\hat{p}_{\text{robust}}$.

**Robust Self-Attention Mechanism** We now introduce the robust self-attention mechanism that down-weights atypical samples. We consider the density estimator of the joint distribution and the marginal distribution when using isotropic Gaussian kernel:

$$\hat{p}_{\text{robust}}(\mathbf{v}, \boldsymbol{k}) = \sum_{j\in[N]} \omega_j^{\text{joint}} k_\sigma([\mathbf{v}_j, \boldsymbol{k}_j], [\mathbf{v}, \boldsymbol{k}]), \quad \hat{p}_{\text{robust}}(\boldsymbol{k}) = \sum_{j\in[N]} \omega_j^{\text{marginal}} k_\sigma(\boldsymbol{k}_j, \boldsymbol{k}). \tag{8}$$

Following the non-parametric regression formulation of self-attention in Eq. (3), we obtain the robust self-attention mechanism as

$$\widehat{\boldsymbol{h}}_i = \frac{\sum_{j\in[N]} \mathbf{v}_j \omega_j^{\text{joint}} k_\sigma(\boldsymbol{q}_i - \boldsymbol{k}_j)}{\sum_{j\in[N]} \omega_j^{\text{marginal}} k_\sigma(\boldsymbol{q}_i - \boldsymbol{k}_j)}, \tag{9}$$

where $\omega^{\text{joint}}$ and $\omega^{\text{marginal}}$ are obtained via either alternative updates or the QP solver. We term Transformers whose density from the non-parametric regression formulation of self-attention employs Eq. (5) and Eq. (6) as Transformer-RKDE and Transformer-SPKDE, respectively. Figure 2 presents an example of the application of the attention weight factor during the learning process from image and text data. The derived weight factor can potentially emphasize elements relevant to the class while reducing the influence of detrimental ones. Note that, the computation of $\{\omega_j^{\text{marginal}}\}_{j\in[N]}$

and $\{\omega_j^{\mathrm{joint}}\}_{j \in [N]}$ are separate as $\omega_j^{\mathrm{joint}}$ involves both keys and values vectors. During the empirical evaluation, we concatenate the keys and values along the head dimension to obtain the weights for the joint density $\hat{p}_{\mathrm{robust}}(\mathbf{v}, \boldsymbol{k})$ and only use the key vectors for obtaining the set of weights for the marginal $\hat{p}_{\mathrm{robust}}(\boldsymbol{k})$. In addition, $\omega^{\mathrm{marginal}}, \omega^{\mathrm{joint}} \in \mathbb{R}^{j \times i}$ for $i, j = 1, \dots, N$ are 2-dimensional matrices that include the pairwise weights between each position of the sequence and the rest of the positions. The weights are initialized uniformly across a certain sequence length dimension. For experiments related to language modeling, we can leverage information from the attention mask to initialize the weights on the unmasked part of the sequence.

**Limitations** While constructing attention weight factors proves effective, they necessitate iterative algorithms to calculate the set of weights when computing self-attention at each layer, resulting in increased overall complexity. Moreover, the foundational contamination model for these methods is the classical Huber contamination model (Huber, 2011), which requires assumptions about contamination distributions and parameters that may not be universally applicable, especially in discrete settings. To address these limitations, we propose a novel approach that circumvents these computational constraints while effectively fostering robust self-attention mechanisms.

### 3.2 Robust Self-Attention via Median-of-Means Principle

The Median-of-Means (MoM) principle (Jerrum et al., 1986; Alon et al., 1996) is one other way to construct robust estimators. Rather than taking the average of all the observations, the sample is split into several blocks over which the median is computed. The MoM principle also improved the robustness of KDE and demonstrated its statistical performance under a less restrictive outlier framework (Humbert et al., 2022). More importantly, it can be easily adapted to self-attention. Specifically, we randomly divide the keys $\{\boldsymbol{k}_j\}_{j=1}^N$ into $B$ subsets $I_1, \dots, I_B$ of equal size, namely, $|I_1| = |I_2| = \dots = |I_B| = \mathcal{S}$. Then, the robust estimator of $p(\boldsymbol{k})$ takes the following form:

$$\hat{p}_{\mathrm{robust}}(\boldsymbol{k}) \propto \mathrm{median}\left\{\hat{p}_{\sigma, I_1}(\boldsymbol{k}), \dots, \hat{p}_{\sigma, I_B}(\boldsymbol{k})\right\}, \tag{10}$$

where we define $\hat{p}_{\sigma, I_l}(\boldsymbol{k}) = \frac{1}{\mathcal{S}} \sum_{j \in I_l} k_\sigma(\boldsymbol{k} - \boldsymbol{k}_j)$ for any $l \in [B]$. Similarly, the robust estimator of $p(\mathbf{v}, \boldsymbol{k})$ is as follows:

$$\hat{p}_{\mathrm{robust}}(\mathbf{v}, \boldsymbol{k}) \propto \mathrm{median}\left\{\hat{p}_{\sigma, I_1}(\mathbf{v}, \boldsymbol{k}), \dots, \hat{p}_{\sigma, I_B}(\mathbf{v}, \boldsymbol{k})\right\}, \tag{11}$$

where $\hat{p}_{\sigma, I_l}(\mathbf{v}, \boldsymbol{k}) = \frac{1}{\mathcal{S}} \sum_{j \in I_l} k_\sigma(\mathbf{v} - \mathbf{v}_j) k_\sigma(\boldsymbol{k} - \boldsymbol{k}_j)$ for any $l \in [B]$. We now propose the self-attention mechanism utilizing the median-of-means principle.

**Median-of-Means Self-Attention Mechanism** Given the robust estimators in Eq. (10) and (11), we can consider the following robust estimation of the attention:

$$\widehat{\boldsymbol{h}}_i = \frac{\frac{1}{\mathcal{S}} \sum_{j \in I_l} v_j k_\sigma(\boldsymbol{q}_i - \boldsymbol{k}_j)}{\mathrm{median}\left\{\hat{p}_{\sigma, I_1}(\boldsymbol{q}_i - \boldsymbol{k}), \dots, \hat{p}_{\sigma, I_B}(\boldsymbol{q}_i - \boldsymbol{k})\right\}}, \tag{12}$$

where $I_l$ is the block such that $\hat{p}_\sigma(\boldsymbol{q}_i - \boldsymbol{k})$ achieves its median value in equation (11). In this context, the random subsets apply to input sequences rather than individual data points, distinguishing this approach from stochastic batches. It's worth noting that our proposed attention mechanism assumes that key and query vectors achieve their median on the same block. Consequently, we apply the median block $I_l$ obtained from the denominator into the numerator, rather than considering the median over all potential blocks, which leads to a process that is faster than computing median blocks on both sides. Moreover, the original MoM principle mandates that each subset be non-overlapping, i.e. $I_{l_1} \cap I_{l_2} = \emptyset$ for any $1 \le l_1 \ne l_2 \le B$. However, for structured, high-dimensional data, dividing into non-overlapping blocks may result in the model only gaining a partial perspective of the dataset, leading to sub-optimal performance. Therefore, we construct each subset by sampling with replacement from the original dataset, maintaining the sequential relationship thereafter. In particular, we have found this strategy to be effective in discrete contexts, such as identifying and filtering out aberrant words in a sentence. As illustrated in the bottom right segment of Figure 2, under a word swap attack, the MoM robust attention retains the subsequence that is most relevant to the content, while discarding unhelpful parts. The downside of MoM self-attention is also apparent: since the attention mechanism only accesses a portion of the sequence, it is likely to result in suboptimal performance on *clean datasets*. The simplicity of MoM allows for easy integration with many existing models. We initially demonstrate that the MoM self-attention mechanism can enhance the recent state-of-the-art FourierFormer (Nguyen et al., 2022c). The theoretical explanation for the ability to remove outliers using MoM-Fourier attention can be found in Appendix C.

Table 1: Perplexity (PPL) and negative likelihood loss (NLL) of our methods (lower part) and baselines (upper part) on WikiText-103. The best results are highlighted in bold font and the second best are highlighted in underline. On clean data, Transformer-SPKDE achieves better PPL and NLL than other baselines. Under random swap with outlier words., Transformers with MoM self-attention show much better performance.

| Method (small version) | Clean Data | | Word Swap | |
|---|---|---|---|---|
| | Valid PPL/Loss | Test PPL/Loss | Valid PPL/Loss | Test PPL/Loss |
| Transformer (Vaswani et al., 2017b) | 33.15/3.51 | 34.29/3.54 | 72.28/4.45 | 74.56/4.53 |
| Performer (Choromanski et al., 2021) | 32.35/3.48 | 33.49/3.51 | 71.64/4.42 | 73.48/4.49 |
| Transformer-MGK (Nguyen et al., 2022b) | 32.28/3.47 | 33.21/3.51 | 69.78/4.38 | 71.03/4.41 |
| FourierFormer (Nguyen et al., 2022c) | 31.86/3.44 | 32.85/3.49 | 65.76/4.32 | 68.33/4.36 |
| Transformer-RKDE (Huber) | 31.22/3.42 | 32.29/3.47 | 52.14/3.92 | 55.68/3.99 |
| Transformer-RKDE (Hampel) | 31.24/3.42 | 32.35/3.48 | 55.61/3.98 | 57.92/4.03 |
| Transformer-SPKDE | **31.05/3.41** | **32.18/3.46** | 51.36/3.89 | 54.97/3.96 |
| Transformer-MoM | 33.56/3.52 | 34.68/3.55 | 48.29/3.82 | 52.14/3.92 |
| FourierFormer-MoM | 32.26/3.47 | 33.14/3.50 | **47.66/3.81** | **50.96/3.85** |

## 3.3 Incorporating Robust Self-Attention Mechanisms into Transformers

**Computational Efficiency** The two types of robust attention mechanisms proposed above have their own distinct advantages. To expedite the computation for Transformer-RKDE and obtain the attention weight factor in a more efficient manner, we employ a single-step iteration on the alternative updates to approximate the optimal set of weights. Empirical results indicate that this one-step iteration can produce sufficiently precise results. For Transformer-SPKDE, as the optimal set of weights is acquired via the QP solver, it demands more computation time but yields superior performance on both clean and contaminated data. As an alternative to weight-based methods, Transformer-MoM offers significantly greater efficiency while providing competitive performance, particularly with text data. The complete procedure for computing the attention vector for Transformer-RKDE, Transformer-SPKDE, and Transformer-MoM is detailed in Algorithm 1.

**Training and Inference** We incorporate our robust attention mechanisms into both training and inference stages of Transformers. Given the uncertainty about data cleanliness, there's a possibility of encountering contaminated samples at either stage. Therefore, it is worthwhile to defend against outliers throughout the entire process. In the training context, our methods modify the computation of attention vectors across each Transformer layer, making them less susceptible to contamination from outliers. This entire process remains nonparametric, with no introduction of additional model parameters. However, the resulting attention vectors, whether shaped by re-weighting or the median-of-means principle, diverge from those generated by the standard softmax attention. This distinction influences the model parameters learned during training. During inference, the test data undergoes a similar procedure to yield robust attention vectors. This ensures protection against potential outlier-induced disruptions within the test sequence. However, if we assume the availability of a clean training set — where contamination arises solely from distribution shifts, adversarial attacks, or data poisoning during inference — it is sufficient to restrict the application of the robust attention mechanism to just the inference phase. This could considerably reduce the computational time required for robust attention vector calculation during training. In our empirical evaluation, we engaged the robust attention mechanism throughout both phases and recorded the associated computational time during training. We found that on standard datasets like ImageNet-1K, WikiText-103, and the UEA time-series classification, infusing the robust attention led to a drop in training loss. This suggests that training data itself may contain inherent noise or outliers.

## 4 Experimental Results

In this section, we provide empirical validation of the benefits of integrating our proposed robust KDE attention mechanisms (Transformer-RKDE/SPKDE/MoM) into Transformer base models. We compare these with the standard softmax Transformer across multiple datasets representing different modalities. These include language modeling on the WikiText-103 dataset (Merity et al., 2016) (Section 4.1) and image classification on ImageNet (Russakovsky et al., 2015; Deng et al., 2009). Furthermore, we assess performance across multiple robustness benchmarks, namely ImageNet-C (Hendrycks & Dietterich, 2019), ImageNet-A (Hendrycks et al., 2021b), ImageNet-O (Hendrycks et al., 2021b), ImageNet-R (Hendrycks et al., 2021a), and ImageNet-Sketch (Wang et al., 2019) (Section 4.2), as well as UEA time-series classification (Section 4.3). Our proposed robust trans-

Table 2: Top-1 and top-5 accuracy (%) on ImageNet. The best results are highlighted in bold font and the second best are highlighted in underlines. RVT (Mao et al., 2022) and DeiT (Touvron et al., 2021b) achieve better results on clean data; meanwhile, Transformers incorporating robust self-attention hold stronger defense under different adversarial attacks while still achieving competitive performance on the original ImageNet.

| Method | Clean Data | | FGSM | | PGD | | SPSA | |
|---|---|---|---|---|---|---|---|---|
| | Top 1 | Top 5 | Top 1 | Top 5 | Top 1 | Top 5 | Top 1 | Top 5 |
| ViT (Dosovitskiy et al., 2020) | 72.23 | 91.13 | 52.61 | 82.26 | 41.84 | 76.49 | 48.34 | 79.36 |
| DeiT (Touvron et al., 2021b) | 74.32 | 93.72 | 53.24 | 84.07 | 41.72 | 76.43 | 49.56 | 80.14 |
| RVT (Mao et al., 2022) | **74.37** | **93.89** | 53.67 | 84.11 | 43.39 | 77.26 | 51.43 | 80.98 |
| FourierFormer (Nguyen et al., 2022c) | 73.25 | 91.66 | 53.08 | 83.95 | 41.34 | 76.19 | 48.79 | 79.57 |
| ViT-RKDE (Huber) | 72.83 | 91.44 | 55.83 | 85.89 | 44.15 | 79.06 | 52.42 | 82.03 |
| ViT-RKDE (Hampel) | 72.94 | 91.63 | 55.92 | 85.97 | 44.23 | 79.16 | 52.48 | 82.07 |
| ViT-SPKDE | 73.22 | 91.95 | **56.03** | **86.12** | **44.51** | **79.47** | **52.64** | **82.33** |
| ViT-MoM | 71.94 | 91.08 | 55.76 | 85.23 | 43.78 | 78.85 | 49.38 | 80.02 |
| FourierFormer-MoM | 72.58 | 91.34 | 53.25 | 84.12 | 41.38 | 76.41 | 48.82 | 79.68 |

formers are compared with state-of-the-art models, including Performer (Choromanski et al., 2021), MGK (Nguyen et al., 2022a), RVT (Mao et al., 2022), and FourierFormer (Nguyen et al., 2022c). All experiments were conducted on machines with 4 NVIDIA A-100 GPUs. For each experiment, Transformer-RKDE/SPKDE/MoM were compared with other baselines under identical hyperparameter configurations.

## 4.1 Robust Language Modeling

WikiText-103 is a language modeling dataset that contains collection of tokens extracted from good and featured articles from Wikipedia, which is suitable for models that can leverage long-term dependencies. We follow the standard configurations in Merity et al. (2016); Schlag et al. (2021) and splits the training data into $L$-word independent long segments. During evaluation, we process the text sequence using a sliding window of size $L$ and feed into the model with a batch size of 1. The last position of the sliding window is used for computing perplexity except in the first segment, where all positions are evaluated as in Al-Rfou et al. (2019); Schlag et al. (2021).

In our experiments, we utilized the small and medium (shown in Appendix E) language models developed by Schlag et al. (2021). We configured the dimensions of key, value, and query to 128, and set the training and evaluation context length to 256. We contrasted our methods with Performer (Choromanski et al., 2021), Transformer-MGK (Nguyen et al., 2022a) and FourierFormer (Nguyen et al., 2022c), which have demonstrated competitive performance. For self-attention, we allocated 8 heads for our methods and Performer, and 4 for Transformer-MGK. The dimension of the feed-forward layer was set to 2048, with the number of layers established at 16. To prevent numerical instability, we used the `log-sum-exp` trick in equation (3) when calculating the attention probability vector through the Gaussian kernel. We employed similar tactics when computing the attention weight factor of Transformer-RKDE, initially obtaining the weights in `log` space, followed by the `log-sum-exp` trick to compute robust self-attention as outlined in equation (9). For Transformer-MoM, the sampled subset sequences constituted 80% of the length of the original sequence.

Table 1 presents the validation and test perplexity (PPL) for several methods. Both Transformer-RKDE and SPKDE models exhibit an improvement over baseline PPL and NLL on both validation and test sets. However, MoM-based models display slightly higher perplexity, a result of using only a portion of the sequence. When the dataset is subjected to a word swap attack, which randomly substitutes selected keywords with a generic token "$AAA$" during evaluation, our method, particularly MoM-based robust attention, yields significantly better results. This is particularly evident when filtering out infrequent words, where the median trick has proven its effectiveness. We also noticed superior robustness in RKDE/SPKDE-based robust attention compared to other baseline methods that were not protected from the attack. Our implementation of the word swap is based on the publicly available TextAttack code by Morris et al. (2020)[1]. We employed a greedy search method with constraints on stop-word modifications provided by the TextAttack library.

## 4.2 Image Classification under Adversarial Attacks and Data Corruptions

For initial simplicity, we utilize the original ViT (tiny) as our base model. However, our methods are versatile and can be integrated with more advanced base models to enhance their robustness. As our

---

[1]Implementation available at github.com/QData/TextAttack

Table 3: We evaluated the performance of proposed models across multiple robustness benchmarks, using appropriate evaluation metrics for each. In the majority of cases, our methods outperformed the baselines.

| Dataset | ImageNet-C | ImageNet-A | ImageNet-O | ImageNet-R | ImageNet-Sketch |
|---|---|---|---|---|---|
| Metric | mCE↓ | Top-1 Acc↑ | AUPR↑ | Top-1 Err Rate↓ | Top-1 Acc↑ |
| ViT | 71.14 | 0.18 | 18.31 | 96.89 | 37.13 |
| DeiT | 70.26 | 0.73 | 19.56 | 94.23 | 41.68 |
| RVT | 68.57 | **9.45** | 22.14 | 63.12 | 50.07 |
| FourierFormer | 71.07 | 0.69 | 18.47 | 94.15 | 38.36 |
| ViT-RKDE (Huber) | 68.69 | 6.98 | 25.61 | 64.33 | 45.63 |
| ViT-RKDE (Hampel) | 68.55 | 6.34 | 26.14 | 62.16 | 45.76 |
| ViT-SPKDE | **68.34** | 8.29 | 28.42 | **61.39** | **50.13** |
| ViT-MoM | 69.11 | 2.29 | 28.08 | 76.44 | 36.92 |
| FourierFormer-MoM | 70.62 | 2.42 | **28.86** | 74.15 | 39.44 |

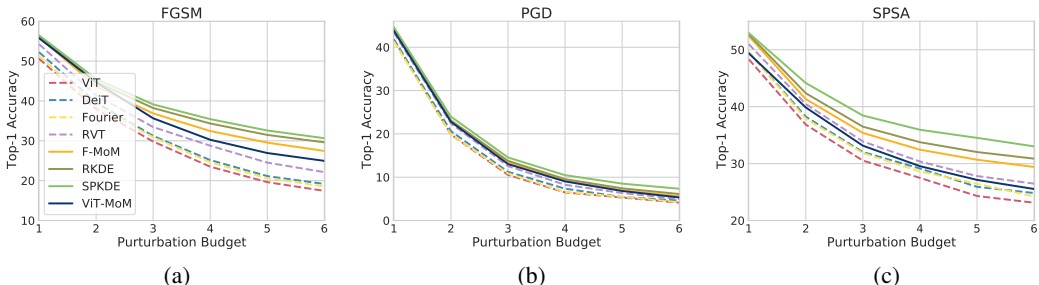

(a)        (b)        (c)

Figure 3: The top-1 classification *accuracy v.s. perturbation budget × 255* curves on ImageNet against three untargeted attack methods under the $l_\infty$ norm. The proposed set of ViT with robust self-attention mechanisms shows stronger defense under all attack methods with different perturbation budgets.

approaches do not alter the model architecture, each model employs $5.7M$ parameters. We have also implemented leading-edge methods, including DeiT with hard distillation (Touvron et al., 2021b), FourierFormer (Nguyen et al., 2022c), and Robust Vision Transformer (RVT) (Mao et al., 2022), as our baselines. It's important to note that, for a fair comparison with RVT, we only incorporated its position-aware attention scaling without further architectural modifications. Consequently, the resulting RVT model comprises approximately $7.2M$ parameters. To evaluate adversarial robustness, we utilized adversarial examples generated by untargeted white-box attacks, which included the single-step attack method FGSM (Goodfellow et al., 2014), multi-step attack method PGD (Madry et al., 2017), and score-based black-box attack method SPSA (Uesato et al., 2018). These attacks were applied to the entire validation set of ImageNet. Each attack distorts the input image with a perturbation budget $\epsilon = 1/255$ under $l_\infty$ norm, while the PGD attack uses 20 steps with a step size of $\alpha = 0.15$. In addition, we assessed our methods on multiple robustness benchmarks, which include images derived from the original ImageNet through algorithmic corruptions or outliers.

Table 2 presents the results under adversarial attack. On clean ImageNet, our performance aligns closely with RVT and DeiT, the leading performers. Notably, our methods outperform RVT under several adversarial attack types, particularly when employing the ViT-SPKDE method. Figure 3 illustrates the relationship between accuracy and perturbation budget across three attack methods. We observe that transformers equipped with robust self-attention mechanisms offer significantly enhanced defense capabilities across different perturbation budgets, with their advantages amplifying as the level of perturbation increases, as expected. We provide more ablation studies in Appendix E that explore dif-

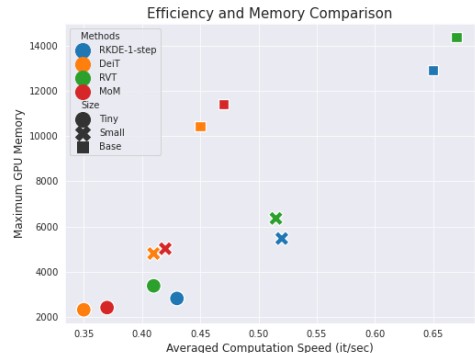

Figure 4: Comparison of averaged computation speed (measured by iteration per second) and maximum GPU memory (measured by the CUDA `max_memory_allocated` function) on ImageNet classification task. The results are measured under the Transformer base models with different capacities: Tiny (5.7M parameters), Small (22M), and Base (86M).

ferent design choices for each proposed robust KDE attention. Table 3 displays the results across multiple robustness benchmarks, employing appropriate evaluation metrics for each. In most instances, the best performance is achieved by our proposed methods, which clearly improve upon existing baselines. Additionally, Figure 4 demonstrates the scalability of our methods as model size increases. We have excluded the SPKDE-based method from this analysis due to its heightened computational demands. The results indicate that both the computation speed and memory increase as model capacity expands for all methods. The MoM-based robust attention closely mirrors the vanilla Transformer, while the RKDE-based robust attention, with one-step approximation, also demonstrates scalability with larger models.

Table 4: A comparison of classification accuracy using the UEA Time Series Classification Archive. Transformers that incorporate our proposed robust attention mechanisms outperform the existing baselines.

| Model/Dataset | Ethanol | Heart | PEMS-SF | Spoken | UWave |
|---|---|---|---|---|---|
| Transformer | 33.70 | 75.77 | 82.66 | 99.33 | 84.45 |
| FourierFormer | 36.12 | **76.42** | 86.70 | 99.00 | 86.66 |
| Transformer-RKDE (Huber) | 34.72 | 75.84 | 84.28 | 99.28 | 86.49 |
| Transformer-SPKDE | 36.09 | 76.29 | 86.02 | **99.36** | **88.14** |
| Transformer-MoM | 38.41 | 73.24 | 86.75 | 97.64 | 82.97 |
| FourierFormer-MoM | **39.89** | 74.11 | **87.63** | 98.12 | 85.43 |

### 4.3 UEA Time Series Classification

Lastly, we conducted experiments using five datasets from the UEA Time-Series Classification Archive (Bagnall et al., 2018), and compared the outcomes across various methodologies (Table 4). The baseline implementation and datasets were adapted from Wu et al. (2022). Our findings indicate that our proposed approaches can notably enhance classification accuracy.

## 5 Conclusion and Future Work

In this work, we explored the link between the dot-product self-attention mechanism and non-parametric kernel regression. This led to the development of a family of fortified transformers, which leverage robust KDE as an alternative to dot-product attention, mitigating the impacts of contaminated samples. We proposed two variants of robust self-attention mechanisms designed to either down-weight or filter out potential corrupted data, both of which can be seamlessly integrated into commonly used transformer models. As our ongoing effort, we are exploring more efficient techniques for estimating the weight set for robust KDE in extremely large models, and incorporating regularization strategies to mitigate the instability of kernel regression with outliers.

## 6 Acknowledgment

Xing Han and Joydeep Ghosh acknowledge support from Intuit Inc. Nhat Ho acknowledges support from the NSF IFML 2019844 and the NSF AI Institute for Foundations of Machine Learning.

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

# Supplementary Material of "Designing Robust Transformers using Robust Kernel Density Estimation"

## A   The Non-parametric Regression Perspective of Self-Attention

Given an input sequence $\boldsymbol{X} = [\boldsymbol{x}_1, \ldots, \boldsymbol{x}_N]^\top \in \mathbb{R}^{N \times D_x}$ of $N$ feature vectors, the self-attention mechanism transforms it into another sequence $\mathbf{H} := [\boldsymbol{h}_1, \cdots, \boldsymbol{h}_N]^\top \in \mathbb{R}^{N \times D_v}$ as follows:

$$\boldsymbol{h}_i = \sum_{j \in [N]} \mathrm{softmax}\Big(\frac{\boldsymbol{q}_i^\top \boldsymbol{k}_j}{\sqrt{D}}\Big) \boldsymbol{v}_j, \text{ for } i = 1, \ldots, N. \tag{13}$$

The vectors $\boldsymbol{q}_i$, $\boldsymbol{k}_j$ and $\boldsymbol{v}_j$ are the query, key and value vectors, respectively. They are computed as follows:

$$[\boldsymbol{q}_1, \boldsymbol{q}_2, \ldots, \boldsymbol{q}_N]^\top := \boldsymbol{Q} = \boldsymbol{X}\boldsymbol{W}_Q^\top \in \mathbb{R}^{N \times D},$$
$$[\boldsymbol{k}_1, \boldsymbol{k}_2, \ldots, \boldsymbol{k}_N]^\top := \boldsymbol{K} = \boldsymbol{X}\boldsymbol{W}_K^\top \in \mathbb{R}^{N \times D}, \tag{14}$$
$$[\boldsymbol{v}_1, \boldsymbol{v}_2, \ldots, \boldsymbol{v}_N]^\top := \boldsymbol{V} = \boldsymbol{X}\boldsymbol{W}_V^\top \in \mathbb{R}^{N \times D_v},$$

where $\boldsymbol{W}_Q, \boldsymbol{W}_K \in \mathbb{R}^{D \times D_x}$, $\boldsymbol{W}_V \in \mathbb{R}^{D_v \times D_x}$ are the weight matrices. Equation (13) can be written in the following equivalent matrix form:

$$\mathbf{H} = \mathrm{softmax}\Big(\frac{\boldsymbol{Q}\boldsymbol{K}^\top}{\sqrt{D}}\Big)\boldsymbol{V}, \tag{15}$$

where the softmax function is applied to each row of the matrix $(\boldsymbol{Q}\boldsymbol{K}^\top)/\sqrt{D}$. Equation (15) is also called the "softmax attention". Assume we have the key and value vectors $\{\boldsymbol{k}_j, \mathbf{v}_j\}_{j \in [N]}$ that is collected from the data generating process

$$\mathbf{v} = f(\boldsymbol{k}) + \varepsilon, \tag{16}$$

where $\varepsilon$ is some noise vectors with $\mathbb{E}[\varepsilon] = 0$, and $f$ is the function that we want to estimate. If $\{\boldsymbol{k}_j\}_{j \in [N]}$ are i.i.d. samples from the distribution $p(\boldsymbol{k})$, and $p(\mathbf{v}, \boldsymbol{k})$ is the joint distribution of $(\mathbf{v}, \boldsymbol{k})$ defined by equation (16), we have

$$f(\boldsymbol{k}) = \mathbb{E}[\mathbf{v}|\boldsymbol{k}] = \int_{\mathbb{R}^D} \mathbf{v} \cdot p(\mathbf{v}|\boldsymbol{k}) d\mathbf{v} = \int_{\mathbb{R}^D} \frac{\mathbf{v} \cdot p(\mathbf{v}, \boldsymbol{k})}{p(\boldsymbol{k})} d\mathbf{v}, \tag{17}$$

We need to obtain estimations for both the joint density function $p(\mathbf{v}, \boldsymbol{k})$ and the marginal density function $p(\boldsymbol{k})$ to obtain function $f$, one popular approach is the kernel density estimation:

$$\hat{p}_\sigma(\mathbf{v}, \boldsymbol{k}) = \frac{1}{N} \sum_{j \in [N]} k_\sigma\left([\mathbf{v}, \boldsymbol{k}] - [\mathbf{v}_j, \boldsymbol{k}_j]\right) \tag{18}$$

$$\hat{p}_\sigma(\boldsymbol{k}) = \frac{1}{N} \sum_{j \in [N]} k_\sigma(\boldsymbol{k} - \boldsymbol{k}_j), \tag{19}$$

where $[\mathbf{v}, \boldsymbol{k}]$ denotes the concatenation of $\mathbf{v}$ and $\boldsymbol{k}$. $k_\sigma$ could be isotropic Gaussian kernel: $k_\sigma(\boldsymbol{x} - \boldsymbol{x}') = \exp\left(-\|\boldsymbol{x} - \boldsymbol{x}'\|^2/(2\sigma^2)\right)$, we have

$$\hat{p}_\sigma(\mathbf{v}, \boldsymbol{k}) = \frac{1}{N} \sum_{j \in [N]} k_\sigma(\mathbf{v} - \mathbf{v}_j) k_\sigma(\boldsymbol{k} - \boldsymbol{k}_j). \tag{20}$$

Combining equations (19), (20), and (17), we obtain the NW estimator of the function $f$ as

$$\widehat{f}_\sigma(\boldsymbol{k}) = \int_{\mathbb{R}^D} \frac{\mathbf{v} \cdot \hat{p}_\sigma(\mathbf{v}, \boldsymbol{k})}{\hat{p}_\sigma(\boldsymbol{k})} d\mathbf{v} \tag{21}$$

$$= \int_{\mathbb{R}^D} \frac{\mathbf{v} \cdot \sum_{j \in [N]} k_\sigma(\mathbf{v} - \mathbf{v}_j) k_\sigma(\boldsymbol{k} - \boldsymbol{k}_j)}{\sum_{j \in [N]} k_\sigma(\boldsymbol{k} - \boldsymbol{k}_j)} d\mathbf{v}$$

$$= \frac{\sum_{j \in [N]} k_\sigma(\boldsymbol{k} - \boldsymbol{k}_j) \int \mathbf{v} \cdot k_\sigma(\mathbf{v} - \mathbf{v}_j) d\mathbf{v}}{\sum_{j \in [N]} k_\sigma(\boldsymbol{k} - \boldsymbol{k}_j)}$$

$$= \frac{\sum_{j \in [N]} \mathbf{v}_j k_\sigma(\boldsymbol{k} - \boldsymbol{k}_j)}{\sum_{j \in [N]} k_\sigma(\boldsymbol{k} - \boldsymbol{k}_j)}. \tag{22}$$

Now we show how the self-attention mechanism is related to the NW estimator. If the keys $\{\boldsymbol{k}_j\}_{j\in[N]}$ are normalized

$$
\begin{aligned}
\widehat{f}_\sigma(\boldsymbol{q}) &= \frac{\sum_{j\in[N]} \mathbf{v}_j \exp\left(-\|\boldsymbol{q}-\boldsymbol{k}_j\|^2/2\sigma^2\right)}{\sum_{j\in[N]} \exp\left(-\|\boldsymbol{q}-\boldsymbol{k}_j\|^2/2\sigma^2\right)} \\
&= \frac{\sum_{j\in[N]} \mathbf{v}_j \exp\left[-\left(\|\boldsymbol{q}\|^2+\|\boldsymbol{k}_j\|^2\right)/2\sigma^2\right] \exp\left(\boldsymbol{q}^\top\boldsymbol{k}_j/\sigma^2\right)}{\sum_{j\in[N]} \exp\left[-\left(\|\boldsymbol{q}\|^2+\|\boldsymbol{k}_j\|^2\right)/2\sigma^2\right] \exp\left(\boldsymbol{q}^\top\boldsymbol{k}_j/\sigma^2\right)} \\
&= \sum_{j\in[N]} \frac{\exp\left(\boldsymbol{q}^\top\boldsymbol{k}_j/\sigma^2\right)}{\sum_{j\in[N]} \exp(\boldsymbol{q}^\top\boldsymbol{k}_j/\sigma^2)} \mathbf{v}_j \\
&= \sum_{j\in[N]} \mathrm{softmax}\left(\boldsymbol{q}^\top\boldsymbol{k}_j/\sigma^2\right)\mathbf{v}_j.
\end{aligned}
\tag{23}
$$

Then estimating the softmax attention is equivalent to estimating $\widehat{f}_\sigma(\boldsymbol{q})$.

# B   Details on Leveraging Robust KDE on Transformers

For simplicity, we use the Huber loss function as the demonstrating example, which is defined as follows:

$$
\rho(x) := \left\{ \begin{array}{ll} x^2/2, & 0 \le x \le a \\ ax - a^2/2, & a < x, \end{array} \right.
\tag{24}
$$

where $a$ is a constant. The solution to this robust regression problem has the following form:

**Proposition 1.** *Assume the robust loss function $\rho$ is non-decreasing in $[0,\infty]$, $\rho(0) = 0$ and $\lim_{x\to 0} \frac{\rho(x)}{x} = 0$. Define $\psi(x) := \frac{\rho'(x)}{x}$ and assume $\psi(0) = \lim_{x\to 0} \frac{\rho'(x)}{x}$ exists and finite. Then the optimal $\hat{p}_{robust}$ can be written as*

$$
\hat{p}_{robust} = \sum_{j\in[N]} \omega_j k_\sigma(\boldsymbol{x}_j, \cdot),
$$

*where $\omega = (\omega_1, \cdots, \omega_N) \in \Delta_N$, with each $\omega_j \propto \psi\left(\|k_\sigma(\boldsymbol{x}_j, \cdot) - \hat{p}_{robust}\|_{\mathcal{H}_{k_\sigma}}\right)$. Here $\Delta_n$ denotes the $n$-dimensional probability simplex.*

*Proof.* The proof of Proposition 1 is mainly adapted from the proof in Kim & Scott (2012). Here, we provide proof of completeness. For any $p \in \mathcal{H}_{k_\sigma}$, we denote

$$
J(p) = \frac{1}{N} \sum_{j\in[N]} \rho\left(\|k_\sigma(\boldsymbol{x}_j, \cdot) - p\|_{\mathcal{H}_{k_\sigma}}\right).
$$

Then we have the following lemma regarding the Gateaux differential of $J$ and a necessary condition for $\hat{p}_{robust}$ to be optimal solution of the robust loss objective function in equation (5).

**Lemma 1.** *Given the assumptions on the robust loss function $\rho$ in Proposition 1, the Gateaux differential of $J$ at $p \in \mathcal{H}_{k_\sigma}$ with incremental $h \in \mathcal{H}_{k_\sigma}$, defined as $\delta J(p; h)$, is*

$$
\delta J(p; h) := \lim_{\tau\to 0} \frac{J(p+\tau h) - J(p)}{\tau} = -\langle V(p), h\rangle_{\mathcal{H}_{k_\sigma}},
$$

*where the function $V : \mathcal{H}_{k_\sigma} \to \mathcal{H}_{k_\sigma}$ is defined as:*

$$
V(p) = \frac{1}{N} \sum_{j\in[N]} \psi\left(\|k_\sigma(\boldsymbol{x}_j, \cdot) - p\|_{\mathcal{H}_{k_\sigma}}\right)(k_\sigma(\boldsymbol{x}_j, \cdot) - p).
$$

*A necessary condition for $\hat{p}_{robust}$ is $V(\hat{p}_{robust}) = 0$.*

The proof of Lemma 1 can be found in Lemma 1 of Kim & Scott (2012). Based on the necessary condition for $\hat{p}_{\text{robust}}$ in Lemma 1, i.e., $V(\hat{p}_{\text{robust}}) = 0$, we have

$$\frac{1}{N} \sum_{j \in [N]} \psi \left( \|k_\sigma(\boldsymbol{x}_j, \cdot) - \hat{p}_{\text{robust}}\|_{\mathcal{H}_{k_\sigma}} \right) (k_\sigma(\boldsymbol{x}_j, \cdot) - \hat{p}_{\text{robust}}) = 0.$$

Direct algebra indicates that $\hat{p}_{\text{robust}} = \sum_{j \in [N]} \omega_j k_\sigma(\boldsymbol{x}_j, \cdot)$ where $\omega = (\omega_1, \cdots, \omega_N) \in \Delta_N$, and $\omega_j \propto \psi \left( \|k_\sigma(\boldsymbol{x}_j, \cdot) - \hat{p}_{\text{robust}}\|_{\mathcal{H}_{k_\sigma}} \right)$. As a consequence, we obtain the conclusion of the proposition. $\square$

For the Huber loss function, we have that

$$\psi(x) := \begin{cases} 1, & 0 \le x \le a \\ a/x, & a < x. \end{cases}$$

Hence, when the error $\|k_\sigma(\boldsymbol{x}_j, \cdot), \cdot - \hat{p}_{\text{robust}}\|_{\mathcal{H}_{k_\sigma}}$ is over the threshold $a$, the final estimator will down-weight the importance of $k_\sigma(\boldsymbol{x}_j, \cdot)$. This is in sharp contrast with the standard KDE method, which will assign uniform weights to all of the $k_\sigma(\boldsymbol{x}_j, \cdot)$. As we mentioned in the main paper, the estimator provided in Proposition 1 is circularly defined, as $\hat{p}_{\text{robust}}$ is defined via $\omega$, and $\omega$ depends on $\hat{p}_{\text{robust}}$. Such an issue can be addressed by estimating $\omega$ with an iterative algorithm termed as kernelized iteratively re-weighted least-squares (KIRWLS). The algorithm starts with randomly initialized $\omega^{(0)} \in \Delta_n$, and perform the following iterative updates between two steps:

$$\hat{p}_{\text{robust}}^{(k)} = \sum_{j \in [N]} \omega_i^{(k-1)} k_\sigma(\boldsymbol{x}_j, \cdot), \quad \omega_j^{(k)} = \frac{\psi \left( \left\| k_\sigma(\boldsymbol{x}_j, \cdot) - \hat{p}_{\text{robust}}^{(k)} \right\|_{\mathcal{H}_{k_\sigma}} \right)}{\sum_{j \in [N]} \psi \left( \left\| k_\sigma(\boldsymbol{x}_j, \cdot) - \hat{p}_{\text{robust}}^{(k)} \right\|_{\mathcal{H}_{k_\sigma}} \right)}. \tag{25}$$

Note that, the optimal $\hat{p}_{\text{robust}}$ is the fixed point of this iterative update, and the KIRWLS algorithm converges under standard regularity conditions. Furthermore, one can directly compute the term $\left\| k_\sigma(\boldsymbol{x}_j, \cdot) - \hat{p}_{\text{robust}}^{(k)} \right\|_{\mathcal{H}_{k_\sigma}}$ via the reproducing property:

$$\left\| k_\sigma(\boldsymbol{x}_j, \cdot) - \hat{p}_{\text{robust}}^{(k)} \right\|_{\mathcal{H}_{k_\sigma}}^2 = -2 \sum_{m \in [N]} \omega_m^{(k-1)} k_\sigma(\boldsymbol{x}_m, \boldsymbol{x}_j) + k_\sigma(\boldsymbol{x}_j, \boldsymbol{x}_j)$$
$$+ \sum_{m \in [N], n \in [N]} \omega_m^{(k-1)} \omega_n^{(k-1)} k_\sigma(\boldsymbol{x}_m, \boldsymbol{x}_n). \tag{26}$$

Therefore, the weights can be updated without mapping the data to the Hilbert space.

## C    Fourier Attention with Median of Means

We introduce the Fourier Attention coupled with the Median of Means (MoM) principle and show how this is robust to outliers. For any given function $\phi : \mathbb{R} \to \mathbb{R}$ and radius $R$, we randomly divide the keys $\{\boldsymbol{k}_i\}_{i \in [N]}$ into $B$ subsets $I_1, \ldots, I_B$ of equal size where $|I_1| = |I_2| = \cdots = |I_B| = \mathcal{S}$. Define $\hat{p}_{R,I_m}(\boldsymbol{q}_l) = \frac{1}{\mathcal{S}} \sum_{i \in I_m} \prod_{j=1}^D \phi(\frac{\sin(R(q_{lj} - k_{ij}))}{R(q_{lj} - k_{ij})})$, then the MoM Fourier attention is defined as

$$\hat{\mathbf{h}}_l = \frac{\frac{1}{\mathcal{S}} \sum_{i \in I_m} \mathbf{v}_i \prod_{j=1}^D \phi(\frac{\sin(R(q_{lj} - k_{ij}))}{R(q_{ij} - k_{lj})})}{\text{median}\{\hat{p}_{R,I_1}(\boldsymbol{q}_l), \ldots, \hat{p}_{R,I_B}(\boldsymbol{q}_l)\}}, \tag{27}$$

where $I_m$ is the block such that $\hat{p}_R(\boldsymbol{q}_l, \boldsymbol{k})$ achieves its median value. To shed light into the robustness of Transformers that use Eq. (27) as the attention mechanism, we demonstrate that the estimator $\hat{p}_R(\boldsymbol{q}) = \text{median}\{\hat{p}_{R,I_1}(\boldsymbol{q}), \ldots, \hat{p}_{R,I_B}(\boldsymbol{q})\}$ is a robust estimator of the density function $p(\boldsymbol{q})$ of the keys. We first introduce a few notations that are useful for stating this result. Denote $\mathcal{C} = \{1 \le i \le N : \boldsymbol{k}_i \text{ is clean}\}$ and $\mathcal{O} = \{1 \le i \le N : \boldsymbol{k}_i \text{ is outlier}\}$. Then, we have $\mathcal{C} \cap \mathcal{O} = \emptyset$ and $\mathcal{C} \cup \mathcal{O} = \{1, 2, \ldots, N\}$. The following result establishes a high probability upper bound on the sup-norm between $\hat{p}_R(\boldsymbol{q})$ and $p(\boldsymbol{q})$.

**Theorem 1.** *Assume that the function $\phi$ satisfies $\int \phi(\sin(z)/z)z^j\,dz = 0$ for all $1 \leq j \leq m$ and $\int |\phi(\sin(z)/z)||z|^{m+1}dz < \infty$ for some $m \in \mathbb{N}$. Furthermore, the density function $p(\boldsymbol{q})$ satisfies $\sup_{\boldsymbol{q}} |p(\boldsymbol{q})| < \infty$. The number of blocks $B$ and the number of outliers $|\mathcal{O}|$ are such that $B > (2+\delta)|\mathcal{O}|$ where $\delta$ is the failure probability. Then, with $\Delta = \frac{1}{2+\delta} - \frac{|\mathcal{O}|}{B}$ for the radius $R$ sufficiently large and $\delta$ sufficiently small, with probability at least $1 - \exp(-2\Delta^2 B)$ we find that*

$$\|\hat{p}_R - p\|_\infty \leq C(\frac{1}{R^{m+1}} + \sqrt{\frac{BR^D \log R \log(2/\delta)}{N}})$$

*where $C$ is some universal constant.*

*Remark* 1. The result of Theorem 1 indicates by choosing $R = \mathcal{O}(N^{-\frac{1}{2(m+1)+D}})$, the rate of $\hat{p}_R$ to $p$ under the supremum norm is $\mathcal{O}(N^{-\frac{m+1}{2(m+1)+D}})$. With that choice of $R$, when $N$ approaches infinity, the MoM estimator $\hat{p}_R$ is a consistent estimator of the clean distribution $p$ of the keys. This confirms the validity of using $\hat{p}_R$ to robustify $p$ and similarly the usage of MoM Fourier attention Eq. (27) as a robust attention for Transformers.

*Proof.* From the formulation of the MoM estimator $\widehat{p}_R(\boldsymbol{q})$, we obtain the following inequality

$$\{\sup_{\boldsymbol{q}} |\hat{p}_R(\boldsymbol{q}) - p(\boldsymbol{q})| \geq \epsilon\} \subset \{\sup_{\boldsymbol{q}} \sum_{b=1}^{B} \mathbf{1}_{\{|\hat{p}_{R,I_b}(\boldsymbol{q})-p(\boldsymbol{q})|\geq\epsilon\}} \geq \frac{B}{2}\}$$

This bound indicates that to bound $\mathbb{P}(\|\hat{p}_R(\boldsymbol{q}) - p(\boldsymbol{q})\|_\infty \geq \epsilon)$, it is sufficient to bound $\mathbb{P}(\{\sup_{\boldsymbol{q}} \sum_{b=1}^{B} \mathbf{1}_{\{|\hat{p}_{R,I_b}(\boldsymbol{q})-p(\boldsymbol{q})|\geq\epsilon\}} \geq \frac{B}{2}\})$. Indeed, for each $1 \leq b \leq B$, we find that

$$\mathbf{1}_{\{|\hat{p}_{R,I_b}(\boldsymbol{q})-p(\boldsymbol{q})|\geq\epsilon\}} \leq \mathbf{1}_{\{\sup_{\boldsymbol{q}}\{|\hat{p}_{R,I_b}(\boldsymbol{q})-p(\boldsymbol{q})|\geq\epsilon\}}.$$

Therefore, we have

$$\sum_{b=1}^{B} \mathbf{1}_{\{|\hat{p}_{R,I_b}(\boldsymbol{q})-p(\boldsymbol{q})|\geq\epsilon\}} \leq \sum_{b=1}^{B} \mathbf{1}_{\{\sup_{\boldsymbol{q}}\{|\hat{p}_{R,I_b}(\boldsymbol{q})-p(\boldsymbol{q})|\geq\epsilon\}},$$

which leads to $\sup_{\boldsymbol{q}} \sum_{b=1}^{B} \mathbf{1}_{\{|\hat{p}_{R,I_b}(\boldsymbol{q})-p(\boldsymbol{q})|\geq\epsilon\}} \leq \sum_{b=1}^{B} \mathbf{1}_{\{\sup_{\boldsymbol{q}}\{|\hat{p}_{R,I_b}(\boldsymbol{q})-p(\boldsymbol{q})|\geq\epsilon\}}$. This inequality shows that

$$\mathbb{P}(\{\sup_{\boldsymbol{q}} \sum_{b=1}^{B} \mathbf{1}_{\{|\hat{p}_{R,I_b}(\boldsymbol{q})-p(\boldsymbol{q})|\geq\epsilon\}} \geq \frac{B}{2}\}) \leq \mathbb{P}(\sum_{b=1}^{B} \mathbf{1}_{\{\sup_{\boldsymbol{q}}\{|\hat{p}_{R,I_b}(\boldsymbol{q})-p(\boldsymbol{q})|\geq\epsilon\}}).$$

To ease the presentation, we denote $W_b = \mathbf{1}_{\{\sup_{\boldsymbol{q}}\{|\hat{p}_{R,I_b}(\boldsymbol{q})-p(\boldsymbol{q})|\geq\epsilon\}}$ and $\mathcal{B} = \{1 \leq b \leq B : I_b \cap \mathcal{O} = \emptyset\}$. Then, the following inequalities hold

$$\begin{aligned}
\sum_{b=1}^{B} \mathbf{1}_{\{\sup_{\boldsymbol{q}}\{|\hat{p}_{R,I_b}(\boldsymbol{q})-p(\boldsymbol{q})|\geq\epsilon\}} &= \sum_{b\in\mathcal{B}} W_b + \sum_{b\in\mathcal{B}^c} W_b \\
&\leq \sum_{b\in\mathcal{B}} W_b + |\mathcal{O}| \\
&\leq \sum_{b\in\mathcal{B}} (W_b - \mathbb{E}[W_b]) + B \cdot \mathbb{P}(\sup_{\boldsymbol{q}} |\hat{p}_{R,I_1}(\boldsymbol{q}) - p(\boldsymbol{q})| > \epsilon) + |\mathcal{O}|,
\end{aligned}$$

where we assume without loss of generality that $1 \in \mathcal{B}$, which is possible due to the assumption that $B > (2+\delta)|\mathcal{O}|$. We now prove the following uniform concentration bound:

**Lemma 2.** *Assume that $\phi(z) \leq C$ for all $|z| \leq 1$ for some universal constant $C$. We have*

$$\mathbb{P}(\sup_{\boldsymbol{q}} |\hat{p}_{R,I_1}(\boldsymbol{q}) - p(\boldsymbol{q})| \geq C(\frac{1}{R^{m+1}} + \sqrt{\frac{R^D \log R \log(2/\delta)}{|I_1|}})) \leq \delta.$$

*Proof of Lemma 2.* By the triangle inequality, we have

$$\sup_{\boldsymbol{q}} |\hat{p}_{R,I_1}(\boldsymbol{q}) - p(\boldsymbol{q})| \leq \sup_{\boldsymbol{q}} |\hat{p}_{R,I_1}(\boldsymbol{q}) - \mathbb{E}[\hat{p}_{R,I_1}(\boldsymbol{q})]| + \sup_{\boldsymbol{q}} |\mathbb{E}[\hat{p}_{R,I_1}(\boldsymbol{q})] - p(\boldsymbol{q})|.$$

To bound $\sup_{\boldsymbol{q}} |\hat{p}_{R,I_1}(\boldsymbol{q}) - \mathbb{E}[\hat{p}_{R,I_1}(\boldsymbol{q})]|$, we use Bernstein's inequality along with the bracketing entropy under $\mathbb{L}_1$ norm in the space of queries $\boldsymbol{q}$. In particular, we denote $Y_i = \frac{R^D}{A^D} \prod_{j=1}^{D} \phi(\frac{\sin(R(q_j - k_{ij}))}{R(q_j - k_{ij})})$ where $A = \int_{\mathbb{R}} \phi(\frac{\sin(z)}{z}) dz$ for all $i \in I_1$. Since $\sin(R(q_j - k_{ij}))/(R(q_j - k_{ij})) \leq 1$ for all $1 \leq j \leq D$, we obtain that $|Y_i| \leq C^D R^D / A^D$ where $C$ is the constant such that $\phi(z) \leq C$ when $|z| \leq 1$. Furthermore, $E[|Y_i|]$

By choose $\epsilon = C(\frac{1}{R^{m+1}} + \sqrt{\frac{R^D \log R \log(2/\delta)}{|I_1|}})$, then we find that

$$\mathbb{P}(\sup_{\boldsymbol{q}} |\hat{p}_{R,I_1}(\boldsymbol{q}) - p(\boldsymbol{q})| > \epsilon) \leq \frac{\delta}{2(2 + \delta)}$$

Collecting the above inequalities leads to

$$\mathbb{P}(\{\sup_{\boldsymbol{q}} \sum_{b=1}^{B} \mathbf{1}_{\{|\hat{p}_{R,I_b}(\boldsymbol{q}) - p(\boldsymbol{q})| \geq \epsilon\}} \geq \frac{B}{2}\}) \leq \exp(-2B\Delta^2),$$

where $\Delta = \frac{1}{2+\delta} - \frac{|\mathcal{O}|}{B}$. As a consequence, we obtain the conclusion of the theorem. $\qquad\square$

## D   Dataset Information

**WikiText-103**   The dataset[1] contains around $268K$ words and its training set consists of about $28K$ articles with $103M$ tokens, this corresponds to text blocks of about 3600 words. The validation set and test sets consist of 60 articles with $218K$ and $246K$ tokens respectively.

**ImageNet**   We use the full ImageNet dataset that contains $1.28M$ training images and $50K$ validation images. The model learns to predict the class of the input image among 1000 categories. We report the top-1 and top-5 accuracy on all experiments. The following ImageNet variants are test sets that are used to evaluate model performance.

**ImageNet-C**   For robustness on common image corruptions, we use ImageNet-C (Hendrycks & Dietterich, 2019) which consists of 15 types of algorithmically generated corruptions with five levels of severity. ImageNet-C uses the mean corruption error (mCE) as a metric: the smaller mCE means the more robust the model under corruption.

**ImageNet-A**   This dataset contains real-world adversarially filtered images that fool current ImageNet classifiers. A 200-class subset of the original ImageNet-1K's 1000 classes is selected so that errors among these 200 classes would be considered egregious, which cover most broad categories spanned by ImageNet-1K.

**ImageNet-O**   This dataset contains adversarially filtered examples for ImageNet out-of-distribution detectors. The dataset contains samples from ImageNet-22K but not from ImageNet-1K, where samples that are wrongly classified as an ImageNet-1K class with high confidence by a ResNet-50 are selected. We use AUPR (area under precision-recall) as the evaluation metric.

**ImageNet-R**   This dataset contains various artistic renditions of object classes from the original ImageNet dataset, which is discouraged by the original ImageNet. ImageNet-R contains 30,000 image renditions for 200 ImageNet classes, where a subset of the ImageNet-1K classes is chosen.

**ImageNet-Sketch**   This dataset contains 50,000 images, 50 images for each of the 1000 ImageNet classes. The dataset is constructed with Google Image queries "sketch of xxx", where xxx is the standard class name. The search is only performed within the "black and white" color scheme.

---

[1] www.salesforce.com/products/einstein/ai-research/the-wikitext-dependency-language-modeling-dataset/

# E  Ablation Studies

In this section, we provide additional results and ablation studies that focus on different design choices for the proposed robust KDE attention mechanisms. The detailed experimental settings can be found in the caption of each table.

Table 5:  Perplexity (PPL) and negative likelihood loss (NLL) of our methods (lower part) and baselines (upper part) on WikiText-103 using a medium version of Transformer. The best results are highlighted in bold font and the second best are highlighted in underline. On clean data, Transformer-SPKDE achieves better PPL and NLL than other baselines. Under random swap with outlier words, Transformers with MoM self-attention show much better performance.

| Method (median version) | Clean Data | | Word Swap | |
|---|---|---|---|---|
| | Valid PPL/Loss | Test PPL/Loss | Valid PPL/Loss | Test PPL/Loss |
| Transformer (Vaswani et al., 2017b) | 27.90/3.32 | 29.60/3.37 | 65.36/4.31 | 68.12/4.36 |
| Performer (Choromanski et al., 2021) | 27.34/3.31 | 29.51/3.36 | 64.72/4.30 | 67.43/4.34 |
| Transformer-MGK (Nguyen et al., 2022b) | 27.28/3.31 | 29.24/3.36 | 64.46/4.30 | 67.31/4.33 |
| FourierFormer (Nguyen et al., 2022c) | 26.51/3.29 | 28.01/3.33 | 63.74/4.28 | 65.27/4.31 |
| Transformer-RKDE (Huber) | 26.12/3.28 | 27.89/3.32 | 49.37/3.85 | 51.22/3.89 |
| Transformer-RKDE (Hampel) | 25.87/3.27 | 27.44/3.31 | 48.62/3.83 | 51.03/3.88 |
| Transformer-SPKDE | **25.76/3.27** | **27.35/3.31** | 46.91/3.79 | 49.14/3.84 |
| Transformer-MoM | 28.26/3.34 | 29.98/3.38 | 45.35/3.75 | 47.92/3.81 |
| FourierFormer-MoM | 27.13/3.31 | 29.02/3.36 | **43.23/3.71** | **44.97/3.74** |

Table 6: Test PPL/NLL loss versus the parameter $a$ of Huber loss function defined in Eq. (24) (upper) and Hampel loss function (Kim & Scott, 2012) (lower; we use $2 \times a$ and $3 \times a$ as parameters $b$ and $c$) on original and word-swapped Wiki-103 dataset. The best results are highlighted in bold font and the second best are highlighted in underline. We choose $a = 0.4$ in rest of the experiments.

| Robust Loss Parameter | 0.1 | 0.2 | 0.4 | 0.6 | 0.8 | 1 |
|---|---|---|---|---|---|---|
| Clean Data | 32.92/3.48 | 32.87/3.48 | **32.29/3.47** | 32.38/3.48 | 32.46/3.48 | 32.48/3.48 |
| Word Swap | 55.82/3.99 | 55.97/3.99 | **55.68/3.99** | 56.89/4.01 | 57.26/4.01 | 57.37/4.01 |
| Clean Data | 32.67/3.48 | **32.32/3.48** | 32.35/3.48 | 32.47/3.48 | 32.53/3.48 | 32.58/3.48 |
| Word Swap | 58.02/4.03 | **57.86/4.03** | 57.92/4.03 | 58.24/4.04 | 58.37/4.04 | 58.43/4.04 |

Table 7: Top-1 classification accuracy on ImageNet versus the parameter $a$ of Huber loss function defined in Eq. (24) under different settings. The best results are highlighted in bold font and the second best are highlighted in underline. We choose $a = 0.2$ in rest of the experiments.

| Huber Loss Parameter | 0.1 | 0.2 | 0.4 | 0.6 | 0.8 | 1 |
|---|---|---|---|---|---|---|
| Clean Data | 71.45 | **72.83** | 71.62 | 71.07 | 70.65 | 70.34 |
| FGSM | **56.72** | 55.83 | 55.34 | 54.87 | 54.02 | 52.98 |
| PGD | **46.37** | 44.15 | 43.87 | 43.25 | 42.69 | 41.96 |
| SPSA | 52.38 | **52.42** | 51.69 | 51.34 | 50.97 | 48.22 |
| Imagenet-C | 45.37 | 45.58 | **45.63** | 45.26 | 44.63 | 43.76 |

Table 8: Top-1 classification accuracy on ImageNet versus the parameter $a$ of Hampel loss function defined in Kim & Scott (2012) under different settings. We use $2 \times a$ and $3 \times a$ as parameters $b$ and $c$. The best results are highlighted in bold font and the second best are highlighted in underline. We choose $a = 0.2$ in rest of the experiments.

| Hampel Loss Parameter | 0.1 | 0.2 | 0.4 | 0.6 | 0.8 | 1 |
|---|---|---|---|---|---|---|
| Clean Data | 71.63 | **72.94** | 71.84 | 71.23 | 70.87 | 70.41 |
| FGSM | **56.42** | 55.92 | 55.83 | 55.66 | 54.97 | 53.68 |
| PGD | **45.18** | 44.23 | 43.89 | 43.62 | 43.01 | 42.34 |
| SPSA | **52.96** | 52.48 | 52.13 | 51.46 | 50.92 | 50.23 |
| Imagenet-C | 44.76 | 45.61 | 46.04 | **46.13** | 45.82 | 45.31 |

Table 9: Top-1 classification accuracy on ImageNet versus the parameter $\beta$ of SPKDE defined in Eq. (6) under different settings. $\beta = \frac{1}{1-\varepsilon} > 1$, where $\varepsilon$ is the percentage of anomalous samples. A larger $\beta$ indicates a more robust model. The best results are highlighted in bold font and the second best are highlighted in underline. We choose $\beta = 1.4$ in rest of the experiments.

| $\beta$ | 1.05 | 1.2 | 1.4 | 1.6 | 1.8 | 2 |
|---|---|---|---|---|---|---|
| Clean Data | **74.25** | 73.56 | 73.22 | 73.01 | 72.86 | 72.64 |
| FGSM | 53.69 | 55.08 | **56.03** | 55.37 | 54.21 | 53.86 |
| PGD | 42.31 | 43.68 | **44.51** | 44.32 | 44.17 | 43.71 |
| SPSA | 51.29 | 52.02 | 52.64 | **52.84** | 52.16 | 51.39 |
| Imagenet-C | 44.68 | **45.49** | 44.76 | 44.21 | 43.96 | 43.33 |

Table 10: Top-1 classification accuracy on ImageNet versus the number of iterations of the KIRWLS algorithm in Eq. (25) employed in Transformer-RKDE. Since the increased number of iterations does not lead to significant improvements of performance while the computational cost is much higher, we use the single-step iteration of the KIRWLS algorithm in Transformer-RKDE.

| | Huber Loss | | | | Hampel Loss | | | |
|---|---|---|---|---|---|---|---|---|
| Iteration # | 1 | 2 | 3 | 5 | 1 | 2 | 3 | 5 |
| Clean Data | 72.83 | 72.91 | 72.95 | 72.98 | 72.94 | 72.99 | 73.01 | 73.02 |
| FGSM | 55.83 | 55.89 | 55.92 | 55.94 | 55.92 | 55.96 | 55.97 | 55.99 |
| PGD | 44.15 | 44.17 | 44.17 | 44.18 | 44.23 | 44.26 | 44.28 | 44.31 |
| SPSA | 52.42 | 52.44 | 52.45 | 52.45 | 52.48 | 52.53 | 52.55 | 52.56 |
| Imagenet-C | 45.58 | 45.61 | 45.62 | 45.62 | 45.61 | 45.66 | 45.68 | 45.71 |

Table 11: Computation time (measured by seconds per iteration) of baseline methods, Transformer-SPKDE, Transformer-MoM and Transformer-RKDE with different number of KIRWLS iterations. Transformer-SPKDE requires longer time since it directly obtains the optimal set of weights via the QP solver.

| | Iterations of KIRWLS | | | | DeiT | RVT | SPKDE | MoM-KDE |
|---|---|---|---|---|---|---|---|---|
| | 1 | 2 | 3 | 5 | | | | |
| Time (s/it) | 0.43 | 0.51 | 0.68 | 0.84 | 0.35 | 0.41 | 1.45 | 0.37 |

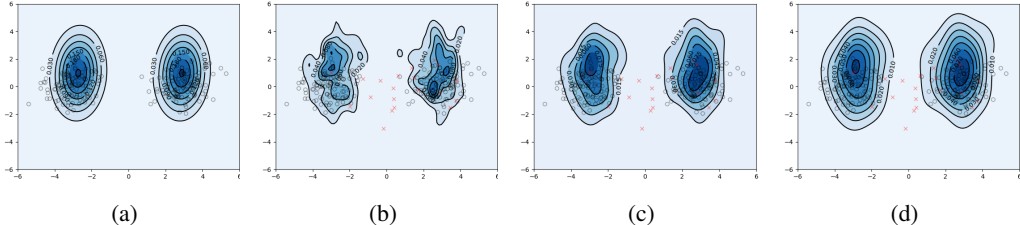

|     |     |     |     |
| --- | --- | --- | --- |
| (a) | (b) | (c) | (d) |

Figure 5: Contour plots of density estimation of the 2-dimensional query vector embedding in an attention layer of the transformer when using (b) KDE (Eq. (4)) and (c) RKDE after one iteration of Eq. (25) with Huber loss (Eq. (24)), (d) KDE with median-of-means principle (Eq. (10)), where (a) is the true density function. We draw 1000 samples (gray circles) from a multivariate normal density and 100 outliers (red cross) from a gamma distribution as the contaminating density. RKDE and KDE with the median-of-means principle can be less affected by contaminated samples when computing self-attention as nonparametric regression.

