# OpenReview forum: "Designing Robust Transformers using Robust Kernel Density Estimation"
_NeurIPS.cc/2023/Conference — NeurIPS 2023 poster_

### Official Review · Reviewer_3B8c · 2023-07-03

**Soundness:** 4 excellent
**Presentation:** 3 good
**Contribution:** 3 good
**Rating:** 7
**Confidence:** 3

**Summary:**

In this work the authors present novel adaptation of a nonstandard self attention module, like those used in transformers. Previous works have demonstrated that the weighting matrix in self attention can be evaluated using a positive kernel rather than the inner product, which ends up looking like a kernel regressor, which itself can be understood through kernel density estimation. In this work the authors propose using robust versions KDEs in place of KDEs for this method. The authors introduce and describe 3 existing versions of robust KDEs for this and apply their method to several experimental settings on a few data types.

On image datasets the authors find that, while their method does not improve the standard performance on clean datasets, the methods improve performance when robustness is advantageous, eg. adversarial samples, dataset shift, etc.. On time series and NLP datasets the authors observe some improvement on the clean data as well as unclean data.

**Strengths:**

The core contribution of this paper is providing a new deep learning methodology that yields substantial performance improvements. The method seems to work on several datatypes and robustness scenarios indicating a fundamental improvement to the general transformer methodology, rather than something that is very application/task/architecture specific. The results also give some insight as to origins of non-robustness of transformer architecture. Its not obvious that key attention values being "heavy-tailed" may make a transformer sensitive to word swapping.

An additional strength of this paper is its novelty. This work is brings methods far outside of what one usually sees in deep learning research, which is also a plus.

Overall the paper is fairly clearly written with a nice overview of a few robust KDEs and introduction to using KDEs for transformers.

**Weaknesses:**

The primary weakness, in my opinion, is that the proposed method is simply swapping out a component of a transformer with a existing robust version of that component, and is somewhat trivial.

**Questions:**

The omega values in the robust KDEs depend on the values k_i (they are functions of $k_i$). During training do the gradients take into account how these omega values will be modified as the functions for k_i are modified? Its a bit opaque to me how the robustness plays into the optimization.

 Could one just train using the standard KDE and then at test time use a robust KDE?

---

> ### Author Rebuttal · Authors · 2023-08-09
>
> Dear Reviewer `3B8c`,
>
> We sincerely appreciate your constructive comments and positive feedback. Such insights play a crucial role in refining our manuscript. We now address the primary concerns you've raised.
>
> **Q: The proposed method is simply swapping out a component of a transformer with an existing robust version of that component.**
>
>
> **A**: While we respectfully contest the statement in question, our study marks an important effort to interpret the robustness issue of Transformers through a nonparametric lens. In this context, we have undertaken significant work to identify effective methodologies capable of tackling this issue.
>
> Initially, our exploration involved linking self-attention to kernel regression problems within the realm of RKHS. This link helped us communicate the primary takeaway of our research, namely, that the robustness concern associated with Transformers can be traced back to the vulnerability of kernel regression problems when confronted with data outliers. However, given the computational constraints and presumptions related to contamination distributions, we were driven to seek alternate methods that could circumvent these issues.
>
> Subsequently, we uncovered a straightforward yet effective approach centered on the principles of the Median-of-Means, which has shown empirical values in discrete contexts. This approach acted as an adjunct methodology for down-weighting outliers in RKHS.
>
> Besides the empirical evidence drawn from a broad spectrum of applications, we argue that each individual component of our study—including the problem formulation, the adaptation of nonparametric methodologies to Transformer settings, and the resolution of practical implementation issues encountered throughout the process—represents a significant, stand-alone contribution to the community.
>
>
> **Q: During training do the gradients take into account how these omega values will be modified as the functions for k_i are modified?**
>
> **A**: From your inquiry, it appears you are primarily referencing the robust self-attention mechanism achieved by down-weighting outliers in RKHS. In our actual implementation, the weight set $\omega$ is kept **separate** from the gradient updates of the Transformer models. Specifically, the `requires_grad` attribute is set to `False` for the weight tensor. The determination of this set of weights is achieved via the iterative methods elaborated upon in our paper. This design choice is motivated by two principal reasons:
>
> (1) Allowing gradient updates for the weight tensors considerably decelerates the training process and, in our observations, doesn't produce any discernible improvements in the end results.
>
> (2) Permitting gradient updates for these tensors could potentially disrupt the gradient flow, making it unstable.
>
> However, this weight set does affect the learned parameters of Transformers. Should there be outliers in the training dataset, this auxiliary weight set introduced would generate attention vectors that differ from those produced by the standard softmax attention mechanism. This, in turn, will influence the model parameters learned during the training phase.
>
>
> **Q: Could one just train using the standard KDE and then at test time use a robust KDE?**
>
> **A**: Yes. Our initial intention is to integrate our robust attention mechanism into both the training and inference stages. Considering that we lack prior knowledge of the data, contaminated samples could potentially emerge during both stages. Consequently, it is worthwhile to safeguard against outliers throughout the process.
>
> In the context of model training, our methods adapt the computation of attention vectors across each Transformer layer, rendering them more resistant to outlier contamination. It's crucial to emphasize that this entire procedure is nonparametric and does not introduce any new model parameters. Nevertheless, the robust attention mechanism, whether it's based on re-weighting or the median-of-means principle, results in attention vectors that are distinct from those generated by the standard softmax attention.
>
> During the inference stage, the test data undergoes the same procedure to generate robust attention vectors, helping to shield against potential contamination from outliers in the test sequence.
>
> However, under the assumption that a clean training set is available, and contaminated samples from distribution shifts, adversarial attacks, or data poisoning only occur during the inference stage, the robust attention mechanism only needs application during inference. This can lead to a significant reduction in the time spent calculating the robust attention vector during training.
>
> Please do not hesitate to let us know if you have any additional questions.
>
> Sincerely,
>
> Authors

---

> > ### Comment · Reviewer_3B8c · 2023-08-14
> >
> > >Besides the empirical evidence drawn from a broad spectrum of applications, we argue that each individual component of our study—including the problem formulation, the adaptation of nonparametric methodologies to Transformer settings, and the resolution of practical implementation issues encountered throughout the process—represents a significant, stand-alone contribution to the community.
> >
> > I suppose I can bump one point, although I feel that the core of the "problem formulation" and initial "adaptation of nonparametric methodologies" was done prior to this work.

---

> > > ### Author Response · Authors · 2023-08-15
> > > **Further Response to Reviewer 3B8c**
> > >
> > > Dear Reviewer `3B8c`,
> > >
> > > Thank you very much for your response and we appreciate your endorsement. We will make sure to add the clarification for the problem formulation and the adaptation of nonparametric methodologies in our future revision.
> > >
> > > Sincerely,
> > >
> > > Authors

---

### Official Review · Reviewer_ATq2 · 2023-07-04

**Soundness:** 3 good
**Presentation:** 3 good
**Contribution:** 4 excellent
**Rating:** 7
**Confidence:** 4

**Summary:**

This paper re-interpret the self-attention mechanism in transformer architectures as a non-parametric kernel density estimator (KDE). Leveraging existing robustness results on classical KDE, the authors develop new variants of transformers; empirical results on a variety of datasets demonstrate how these variants are resistant to adversarial attacks and data contamination.

**Strengths:**

This paper provides a simple yet powerful re-interpretation of the self-attention mechanism through the lens of KDE. This re-interpretation allows to leverage results from robust KDE to design better transformer architectures. Kernel methods are notably amenable to theoretical analysis; therefore, links between transformers and kernels as the one proposed in this paper can help sheding lights on justify the empirical successess of transformer architectures. The paper is in general well-written and its structure is easy to follow.

**Weaknesses:**

Links between Transformers and kernel methods have been proposed in prior literature, see for example [1]. Although an extensive literature review is proposed by the authors, some important references such as [1] are omitted.

**References**
[1] Tsai, Yao-Hung Hubert, et al. "Transformer dissection: a unified understanding of transformer's attention via the lens of kernel." arXiv preprint arXiv:1908.11775 (2019).


**Questions:**

In my view, the paper makes a strong contribution. Could the authors please address my questions below.

- In line 93 when the kernel $k_{\sigma}$ is introduced, it should be specified what set does this kernel act on and the fact that the considered class of kernels is translation invariant $\tilde k_\sigma(x,y) = k_\sigma(x-y)$. For example, equation (2) is ambiguous because the kernels used in the joint density and marginal density estimations are different (indexed on different spaces).
- In line 95-96, it would be good to add an expression to define the mentioned isotropic Gaussian kernel.
- In lines 186-187 it is mentioned that the contruction of attention weight factors necessitate iterative algorithms to calculate the set of weights when computing self-attention at each layer; however, if (sometimes) the weight estimation boils down to a finite dimensional quadratic optimisation problem with linear constraints such as equation (7), can't the weights be determined by simply solving a linear system of equations?
- Kernel regression often involves regularisation; this setup is not considered in the paper. Can the authors comment on the potential role of regularisation?

**Limitations:**

The authors have discussed limitations of their work.

---

> ### Author Rebuttal · Authors · 2023-08-09
>
> Dear Reviewer `ATq2`,
>
> We greatly value your constructive feedback and insights. We will address your suggestions regarding the introduction of kernel functions (questions 1 and 2) and will incorporate any missing citations you've noted in the upcoming version of our manuscript. Thank you for your significant contribution to improving our work. We address your remaining questions below.
>
> **Q: If (sometimes) the weight estimation boils down to a finite-dimensional quadratic optimisation problem with linear constraints such as equation (7), can't the weights be determined by simply solving a linear system of equations?**
>
> **A**: Thank you for raising this point. To the best of our understanding, in a finite-dimensional quadratic optimization problem, one can combine all quadratic terms to transform the problem into a linear equation. However, due to the linear constraint in our formulation, the constraints that result from this transformation remain nonlinear in relation to the quadratic terms. As such, we cannot derive a system of linear equations using this method. We apologize if our interpretation of your question is not accurate and are open to exploring this further with more specific guidance.
>
>
> **Q: Kernel regression often involves regularisation; this setup is not considered in the paper. Can the authors comment on the potential role of regularisation?**
>
> **A**: Thank you for this astute observation. Regularization in kernel regression indeed presents another viable strategy for managing the instability of kernel regression when dealing with data contaminated by errors. Consequently, this perspective may pave the way for developing an alternative robust Transformer model.
>
> However, the reason we didn't address regularized kernel regression in our paper is due to the already comprehensive analysis of three different robust Transformers presented, which rendered the paper quite dense. We therefore thought it prudent to earmark the development of a robust Transformer using regularized kernel regression approaches for future research.
>
> Please do not hesitate to let us know if you have any additional questions.
>
> Sincerely,
>
> Authors

---

> > ### Comment · Reviewer_ATq2 · 2023-08-12
> >
> > I thank the authors for their responses. I keep my rating unchanged for now. I will make a final decision after consultation with other reviewers and AC.

---

> > > ### Author Response · Authors · 2023-08-15
> > > **Further Response to Reviewer ATq2**
> > >
> > > Dear Reviewer `ATq2`,
> > >
> > > Thank you very much for your response and we really appreciate your endorsement!
> > >
> > > Sincerely,
> > >
> > > Authors

---

### Official Review · Reviewer_KTaQ · 2023-07-04

**Soundness:** 3 good
**Presentation:** 2 fair
**Contribution:** 2 fair
**Rating:** 5
**Confidence:** 3

**Summary:**

The paper addresses robustness of transformer architecture in both natural language and vision contexts.
Motivated by classical nonparametric estimation problem, the authors propose a procedure to robustly estimate the self-attention values. While not loosing too much in performance on the original data, the method obtains improvements on corrupted data, with several benchmarks considered. The paper proposes a number of methods to address an important problem of robustness to adversarial contaminations and outliers.

Pros:
- sound experimental results. the proposed method beats sota on a number of benchmarks.
- ablation study is included in the appendix
- comparison of time

Cons:
- I did not understand from reading the introduction whether this method is a post-hoc modification or requires in-processing training
- the theoretical contribution is questionable, I have some questions for Theorem 1 in the appendix, see below
- lack of originality, the strongest part of the paper is experimental results.

minor comments:
- line 84: it is best to refer to Nguyen et al to motivate using this form of regression
- line 84: usually conditional mean is equal to zero E[eps | k] = 0
- line 86: I suggest to make a precise statement that the derivations only hold for Gaussian kernel earlier.
- line 256 "We follow..." correct the grammar in this sentence
- theorem 1 in appendix: the sentence "by adapting Lemma 1 to uniform case". How exactly do you adapt this to uniform case without any loss in the upper bound? I don't think it's that easy to turn a pointwise bound into a uniform one, so I reckon you do not have a proof of the claimed theorem.


Questions:

- do the methods require fine-tunning the parameters?

- how do you choose hyper-parameters, such as number of bins in MoM, cut-off parameter in Huber loss, etc?

- it is completely unclear from the paper whether robust modification only concerns with inference time, or it changes the training as well?

- robust estimators such as MoM are typically robust towards a corruptions of finite amount of data points. Does it mean that you can be robust to one pixel modifications? Or even one 16x16 patch modifications?

- in the experiments, does the hyperparameter choice procedure require the access to contaminated data?

**Strengths:**

-

**Weaknesses:**

-

**Questions:**

-

**Limitations:**

-

---

> ### Author Rebuttal · Authors · 2023-08-09
>
> Dear Reviewer `KTaQ`,
>
> We're really grateful for the thorough and constructive feedback you've provided. Your writing suggestions are highly valuable, and we plan to incorporate all of them in our revised manuscript. We now summarize and address your major questions below.
>
> **Cons 1 & Q3**
>
> **A**: Our intention is to integrate our robust attention mechanism into both the training and inference stages. Considering that we lack prior knowledge of the data, contaminated samples could potentially emerge during both stages. Consequently, it is worthwhile to safeguard against outliers throughout the process.
>
> In the context of model training, our methods adapt the computation of attention vectors across each Transformer layer, rendering them more resistant to outlier contamination. It's crucial to emphasize that this entire procedure is nonparametric and does not introduce any new model parameters. Nevertheless, the robust attention mechanism, whether it's based on re-weighting or the median-of-means principle, results in attention vectors that are distinct from those generated by the standard softmax attention. This, in turn, impacts the model parameters learned during the training procedure.
>
> During the inference stage, the test data undergoes the same procedure to generate robust attention vectors, helping to shield against potential contamination from outliers in the test sequence.
>
> However, under the assumption that a clean training set is available, and contamination from distribution shifts, adversarial attacks, or data poisoning only occurs during the inference stage, the application of the robust attention mechanism may be limited to inference. This approach could considerably reduce the computational time required for robust attention vector calculation during training.
>
> For the empirical evaluation, we applied our robust attention mechanism in both stages and documented the related computation time during training. We found that on standard datasets such as ImageNet-1K, WikiText-103, and the UEA time-series classification, the incorporation of robust attention could also lead to reduced training loss. This suggests that training data itself may contain inherent noise or outliers in many cases.
>
>
> **Cons 3**
>
> **A**: While we respectfully contest the statement in question, our study marks a pioneering effort to interpret the robustness issue of Transformers through a nonparametric lens. In this context, we have undertaken significant work to identify effective methodologies capable of tackling this issue.
>
> Initially, our exploration involved linking self-attention to kernel regression problems within the realm of RKHS. This link helped us communicate the primary takeaway of our research, namely, that the robustness concern associated with Transformers can be traced back to the vulnerability of kernel regression problems when confronted with data outliers. However, given the computational constraints and presumptions related to contamination distributions, we were driven to seek alternate methods that could circumvent these issues.
>
> Subsequently, we uncovered a straightforward yet effective approach centered on the Median-of-Means principle, which has shown empirical values in discrete contexts. This approach acted as an adjunct methodology for down-weighting outliers in RKHS.
>
> Besides the empirical evidence drawn from a broad spectrum of applications, we argue that each individual component of our study—including the problem formulation, the adaptation of nonparametric methodologies to Transformer settings, and the resolution of practical implementation issues encountered throughout the process—represents a significant, stand-alone contribution to the community.
>
>
> **Cons 2**
>
> **A**: Thank you for your comment, and we apologize for any confusion our initial argument may have caused. Indeed, the uniform concentration bound can be adapted from the argument used in Theorem 2 in [1], in conjunction with Lemma 1 in [2]. This is possible when we extend the Fourier Integral Theorem in its generalized form. We will revise the paper to reflect this change.
>
> The core idea behind the uniform concentration bound is the utilization of Bernstein's inequality, along with the bracketing entropy under the $\mathbb{L}_{1}$ norm of the functions in the space of queries, denoted as $q$. The details of this argument can be found within Theorem 2 in [1].
>
> [1] N. Ho and S. Walker. Multivariate Smoothing via the Fourier Integral Theorem and Fourier Kernel. Arxiv preprint, 2020.
>
> [2] T. Nguyen, M. Pham, T. Nguyen, K. Nguyen, S. Osher, N. Ho. Fourierformer:Transformer meets generalized Fourier integral theorem. NeurIPS, 2022.
>
> **Q1 & Q5**
>
> **A**: The methods require fine-tuning and the full set of parameters can be found in the code we've provided. There are several key hyperparameters including
>
> (1) the cut-off parameters in the Huber and Hampel loss function;
>
> (2) the iteration count in the Kernelized Iterative Reweighted Least Squares algorithm;
>
> (3) the scaling factor in the scaled and projected KDE;
>
> (4) the number of partitions or blocks in the Median-of-Means procedure.
>
> Our initial approach involves setting these parameters within a commonly used range, after which we execute a grid search within this predefined space. There is no need to require access to contaminated data during this procedure. The outcomes of this hyperparameter tuning process are detailed in Tables 6 through 10 in the appendix.
>
>
> **Q4**
>
> **A**: As discussed in Section 3.2 of our paper, the robust attention mechanism employing Median-of-Means tends to excel in discrete contexts. When applied to modifications at the pixel or patch level, this approach doesn't demonstrate a clear advantage compared to other methods. This observation is substantiated by the experimental results obtained from Vision Transformers, as presented in Section 4.2 of our paper.
>
> Please let us know if you have additional questions.
>
> Sincerely,
>
> Authors

---

> > ### Comment · Reviewer_KTaQ · 2023-08-13
> >
> > Thank you for your response. Could you please give some further clarifications:
> >
> > > In the context of model training, our methods adapt the computation of attention vectors across each Transformer layer, rendering them more resistant to outlier contamination
> >
> > Could you please give a pointer in the paper, where you describe how it affects the training of the network, in particular, how it affects gradient computation?
> >
> > > The core idea behind the uniform concentration bound is the utilization of Bernstein's inequality, along with the bracketing entropy under the  norm of the functions in the space of queries, denoted as $q$. The details of this argument can be found within Theorem 2 in [1].
> >
> > Could you please clarify, what is dimension of $q$? And what is assumed about the probability density $p$ we are estimating? As far as I know even in the classical case of density estimation, the error bound on $L_{\infty}$ norm depends dramatically on the dimension.

---

> > > ### Author Response · Authors · 2023-08-15
> > > **Further Response to Reviewer KTaQ**
> > >
> > > Dear Reviewer `KTaQ`,
> > >
> > > Thank you for your further comments. We address your questions below:
> > >
> > > **Q: Could you please give a pointer in the paper, where you describe how it affects the training of the network, in particular, how it affects gradient computation?**
> > >
> > > **A:** Thank you for bringing this to our attention. In lines 177 - 188 and 210 - 217, we have elaborated on the method for computing the robust attention mechanism and its integration during the model training phase. Currently, comprehensive details of the model training can be accessed in the provided codebase. In future versions of our manuscript, we will expand on how the robust attention mechanism impacts gradient computation. To directly address your feedback, we've outlined our responses below:
> > > For weight-based methods, the weight set $\omega$ is kept separate from the gradient updates of the Transformer models. Specifically, the `requires_grad` attribute is set to `False` for the weight tensor. The determination of this set of weights is achieved via the iterative methods elaborated upon in our paper. This design choice is motivated by two principal reasons:
> > >
> > > (1) Allowing gradient updates for the weight tensors considerably decelerates the training process and, in our observations, doesn't produce any discernible improvements in the end results.
> > >
> > > (2) Permitting gradient updates for these tensors could potentially disrupt the gradient flow, making it unstable.
> > >
> > > However, this weight set does affect the learned parameters of Transformers. Should there be outliers in the training dataset, this auxiliary weight set introduced would generate attention vectors that differ from those produced by the standard softmax attention mechanism. This, in turn, will influence the model parameters learned during the training phase.
> > >
> > > For the Median-of-Means-based method, determining the median block is non-differentiable. Once this block is identified and employed to compute the robust attention vectors, gradient updates focus solely on a subset of the parameter matrix, derived from the selected indices of the input sequence's median block. As a result, the attention vector, and by extension the model's training process, is minimally impacted by the input sequence's outlier regions, which are typically omitted from the median block.
> > >
> > > In summary, neither of our proposed methods introduces extra differentiable parameters. However, they do affect the model parameters learned during training via the resultant robust attention vectors.
> > >
> > > **Q: Could you please clarify, what is dimension of $q$? And what is assumed about the probability density $p$ we are estimating? As far as I know even in the classical case of density estimation, the error bound on $L_{\infty}$ norm depends dramatically on the dimension.**
> > >
> > > **A:** We apologize for the ambiguity about the assumptions on the theoretical result. We will carefully clarify these assumptions in the revision.
> > >
> > > For your question, the dimension of query $q$ is $D$, which is also the dimension of the key. For the probability density function $p$, we assume that it is differentiable up to order $m + 1$ where $m$ is a constant in the statement of Theorem 1 in Appendix C of the paper.
> > >
> > > From the statement of Theorem 1, by choosing the optimal radius $R$ to balance the bias, which is at the order $O(1/R^{m + 1})$, and the variance, which is at the order of $O(R^{D}/ N)$ (up to some logarithmic function of $R$) in the upper bound for the $L_{\infty}$ norm, the optimal choice of $R$ is $O(N^{1/(D+ 2(m + 1))})$, which leads to the rate $N^{-(m + 1)/(D + 2(m + 1))}$ for the Median-of-Means estimator in Appendix C.
> > >
> > > We note that this rate is standard and similar to the rate of density estimation under the standard Transformer (without any outliers in the data), which is essentially the rate of kernel density estimator with Gaussian kernel.
> > >
> > > Please let us know if you have additional questions.
> > >
> > > Sincerely,
> > >
> > > Authors

---

> > > > ### Comment · Reviewer_KTaQ · 2023-08-15
> > > >
> > > > Thank you for clarification. I think the details of the training procedure are very important part of you contributions and should be made clear and visible. I also encourage you to take a look at the paper below, they also seem to have a sort of EM algorithm for weights. I think it is worth mentioning this paper and adjacent ones if there are any similarities.
> > > >
> > > > Hopkins et al (2020) Robust and Heavy-Tailed Mean Estimation Made Simple, via Regret Minimization
> > > >
> > > > Furthermore, there is a paper that differentiates the MOM objective in the context of clustering, perhaps it should also be cited.
> > > >
> > > > Paul et al (2021) Uniform Concentration Bounds toward a Unified Framework for Robust Clustering
> > > >
> > > > I think the experimental results are very impressive, and I particularly appreciate the fact that you are not using the contaminated samples for validation.
> > > >
> > > > I will retain the score for now due to the lack of clarity. The effect on training procedure should be mentioned as early as in introduction, and the difficulties associated with it should ideally be the addressed in the paper in greater detail.

---

> > > > > ### Author Response · Authors · 2023-08-15
> > > > > **Further Response to Reviewer KTaQ**
> > > > >
> > > > > Dear Reviewer `KTaQ`,
> > > > >
> > > > > Thank you for the valuable insights; they are indeed essential in enhancing our manuscript!
> > > > >
> > > > > We will ensure that our revision includes an in-depth discussion of the works you mentioned, and we'll clarify the details of the training procedure and any practical implementation challenges we have encountered.
> > > > >
> > > > > Sincerely,
> > > > >
> > > > > Authors

---

### Official Review · Reviewer_c4Ug · 2023-07-07

**Soundness:** 3 good
**Presentation:** 3 good
**Contribution:** 3 good
**Rating:** 5
**Confidence:** 4

**Summary:**

This paper introduces fortified transformers that employ robust KDE as an alternative to dot-product attention, self-attention, mitigating the impacts of attacked images. Especially, this paper proposed two variants of robust self-attention mechanisms based on kernel density estimators (KDE). Extensive experimental results demonstrate the effectiveness of the proposed approach on diverse tasks.

**Strengths:**

- This paper is well-written.
- This paper analyzes the characteristics of the proposed methods, especially their limitation.
- Extensive experiments are conducted on diverse tasks with multiple modalities.

**Weaknesses:**

- The paper offers a comprehensive study of several methods, but a dominant strategy does not emerge as the results seem to depend largely on the task at hand, as evidenced in the data tables provided.
- Although the stated limitations are acknowledged, a more in-depth analysis of these shortcomings would enhance the persuasive power.
   - It is acknowledged that RKDE and SPKDE possess computational inefficiencies. Then, how computationally expensive they are than the original ones which are not robust? It would be better to provide the experimental results.
   - Suboptimality is a possible limitation of MoM, then how much the extent of MoM's accessibility of MoM is restricted compared to the originals? How those restricted accessibility affects the performances?


**Questions:**

N/A

---

> ### Author Rebuttal · Authors · 2023-08-09
>
> Dear Reviewer `c4Ug`,
>
> We would like to thank you for your constructive feedback and insightful suggestions! Below we summarize your major questions and address your concerns.
>
> **Q: The paper offers a comprehensive study of several methods, but a dominant strategy does not emerge as the results seem to depend largely on the task at hand, as evidenced in the data tables provided.**
>
> **A**: Thank you for your constructive feedback - it's greatly appreciated and will certainly help us refine and improve our manuscript. Strengthening model robustness has been a major focus in this field for a long time, yet there isn't a one-size-fits-all solution that ensures robustness under all circumstances. This is because the appropriate approach is heavily dependent on several factors, including the nature of the outlier distribution, the data modality, model architectures, and so on.
>
> Our initial aim was to devise a robust attention mechanism that could serve as a standard component for Transformer models, regardless of the task at hand. This was based on the perspectives of nonparametric regression for the self-attention mechanism. However, our experimental findings across various tasks have shown the unique advantages of individual robust attention mechanisms.
>
> It's important to note that our conclusion doesn't revolve around a specific method, but rather the general methodology for designing robust Transformers from the nonparametric regression perspective of the self-attention mechanism.
>
> **Q: It is acknowledged that RKDE and SPKDE possess computational inefficiencies. Then, how computationally expensive they are than the original ones which are not robust? It would be better to provide the experimental results.**
>
> **A**: We've made sure to include a comprehensive comparison of computation times across all the methods featured in our experiments, utilizing the widely-recognized ImageNet-1K dataset. You'll find this in Table 11 in the Appendix. We've also conducted an extra evaluation using the WikiText-103 dataset for the language modeling task; this can be found in the recently added PDF file. In addition, all results of our new investigations can be found in the general announcement.
>
> **Q: Suboptimality is a possible limitation of MoM, then how much the extent of MoM's accessibility of MoM is restricted compared to the originals? How those restricted accessibility affects the performances?**
>
> **A**: Firstly, we'd like to highlight that the suboptimality we observed with the MoM-based robust self-attention is primarily on clean datasets. As we mentioned in the main paper (lines 221 - 223), MoM's attention mechanism uses only a portion of the sequences, which is what leads to this suboptimal performance. However, this approach tends to be more effective in discrete contexts, for instance, when identifying and filtering out aberrant words in a sentence.
>
> In response to your question, the reach of MoM largely hinges on how we define the subsets of the input sequence $I_l, l \in [B]$, such as the size of the subset $\mathcal{S}$, whether the subset overlaps with others, and whether the sequential relationship is retained within the subset. These factors are pivotal to the extent Transformers can recover the full view of the input sequence.
>
> In our experiments, each subset's size, $\mathcal{S}$, is 80% of the original sequence length. We construct each subset by sampling with replacement from the original sequence while maintaining the original sequential relationship. Our empirical results indicate that the performance on clean datasets closely mirrors that of the standard softmax attention, with little impact on performance.
>
> Furthermore, as we've discussed in Section 4.1, the robust attention that uses the Median-of-Means principle can be useful in defending against word swap attacks.
>
> Please do not hesitate to let us know if you have any additional questions.
>
> Sincerely,
>
> Authors

---

> > ### Comment · Reviewer_c4Ug · 2023-08-14
> >
> > Q1. According to your response, the proposed method aims to design a robust attention mechanism that can be applied as task-agnostic. What exact element enables your method to work well against various tasks (model architectures, data modality)? Where is it explicitly designed for your method? What is the theoretical or technical motivation?
> >
> > Q2. Are there any references (other papers) clarifying that existing approaches fail to ensure robustness under diverse circumstances such as data modality, model architectures, and natural distributional shift?
> >
> > Q3. It's not still convinced that the proposed method is regardless of the task and one-size-fits-all solution since you employ a range of robust KDE techniques, not one solution.
> >
> > Q4. In the table you provided in the PDF file, SPKDE requires a 5 times larger cost than baseline, which is inefficient.

---

> > > ### Author Response · Authors · 2023-08-15
> > > **Further Response to Reviewer c4Ug**
> > >
> > > Dear Reviewer `c4Ug`,
> > >
> > > Thank you very much for raising these questions! We address them below:
> > >
> > > **Q1**: The exact element that ensures the robustness of our proposed methods lies in the robust KDE techniques, which are supported by extensive theoretical evidence in various literature. Our investigation connected the self-attention mechanism with nonparametric regression issues, thus offering a pathway to harness the theoretical strengths of robust KDEs. This subsequently fortifies the self-attention mechanism, a pivotal element of Transformer models.
> > >
> > > It's crucial to highlight that there's often a disparity between theory and practical application, leading to potential variations in performance when translating theoretical findings to specific use cases. Given the profound non-linearity of deep neural networks and the intrinsic noise in real-world data sets, it’s possible to have a small misalignment between observed outcomes and the theoretical claims of the robust KDE. This discrepancy has sparked discussions on tailoring versions of robust KDE to suit task-specific applications and addressing the challenges faced during its practical execution. These aspects constitute key facets of our submission.
> > >
> > >
> > > **Q2 & Q3**: We've noted that recent individual robustness methods for Transformers (e.g., [1, 2, 3, 4, 5, 6]) vary in their methodologies, largely due to differences in application domains. To our knowledge, no existing work has proposed a universal solution to tackle the robustness issue of Transformers. Consequently, in our endeavor to craft a robust Transformer model under a generalized framework, we encountered challenges that necessitated subtle customizations in model design to cater to distinct task-specific applications.
> > >
> > > Furthermore, it's worth highlighting that each robust Transformer model we crafted consistently outperformed baseline Transformers in a majority of scenarios. Not only do we provide a nonparametric perspective on Transformer robustness, but we also provide end-users with the flexibility to select from a set of robust Transformers, allowing them to further fine-tune performance based on specific tasks at hand.
> > >
> > > [1] Mahmood et al., 2021. On the Robustness of Vision Transformers to Adversarial Examples, ICCV 2021.
> > >
> > > [2] Mao et al., 2022. Towards Robust Vision Transformer, CVPR 2022.
> > >
> > > [3] Zhou et al., 2022. Understanding the robustness in vision transformers, ICML 2022.
> > >
> > > [4] Yang et al., 2022. TABLEFORMER: Robust Transformer Modeling for Table-Text Encoding, ACL 2022.
> > >
> > > [5] Liu et al., 2021. CrisisBERT: a Robust Transformer for Crisis Classification and Contextual Crisis Embedding, ACM HT 2021.
> > >
> > > [6] Li et al., 2020. RobuTrans: A Robust Transformer-Based Text-to-Speech Model, AAAI 2020.
> > >
> > >
> > > **Q4**: As noted in our paper (lines 231 - 233), while the Transformer-SPKDE does require more computational time compared to other methods, it consistently delivers superior performance across various applications. This is largely due to the optimal weight set obtained through the QP solver. It's a widely acknowledged trade-off that enhancing robustness often comes at the expense of accuracy or computational efficiency. As discussed in Section 5, addressing this challenge will be a critical point in our subsequent research.
> > >
> > > Please let us know if you have more questions.
> > >
> > > Sincerely,
> > >
> > > Authors

---

> > > > ### Comment · Reviewer_c4Ug · 2023-08-17
> > > >
> > > > Thanks for your time and effort in offering responses to my concerns.
> > > >
> > > > I understand the trade-off between enhanced robustness and computational efficiency.
> > > > However, it is not clear for me why the proposed robust attention mechanism works in a task-agnostic way including model architectures and data modality. Therefore, I will maintain my score.

---

> > > > > ### Author Response · Authors · 2023-08-18
> > > > > **Further Response to Reviewer c4Ug**
> > > > >
> > > > > Dear Reviewer `c4Ug`,
> > > > >
> > > > > Thank you for your further comments. Your insights are invaluable in aiding the refinement of our manuscript. We'll be certain to address the concerns you've highlighted in our forthcoming revisions. Below we make a brief response to your query.
> > > > >
> > > > > As indicated in our previous replies, while striving to create a comprehensive solution for the robustness of transformers, we meet with challenges that mandate specific adjustments to the design of the proposed robust attention mechanisms, tailored to particular data modalities. This observation is not only detailed in our manuscript but also supported by our empirical assessments. Furthermore, our method primarily seeks to bolster the robustness of transformers through the lens of the self-attention mechanism, a foundational component in every transformer model, irrespective of its architectural nuances. This means it holds the potential for integration with diverse transformer designs. Given that other fundamental elements of transformers also play pivotal roles in ensuring robustness, as discussed by [1], we recognize that our method might have avenues for further enhancements. We envision addressing this as a part of our future research.
> > > > >
> > > > > [1] Mao et al., 2022. Towards Robust Vision Transformer, CVPR 2022.
> > > > >
> > > > > Thank you once again for your invaluable feedback and the time you dedicated to helping us refine our manuscript!
> > > > >
> > > > > Sincerely,
> > > > >
> > > > > Authors

---

### Official Review · Reviewer_SN7k · 2023-07-11

**Soundness:** 3 good
**Presentation:** 3 good
**Contribution:** 3 good
**Rating:** 6
**Confidence:** 2

**Summary:**

The paper describes and leverages the connection between the self-attention mechanism and KDE estimation to motivate replacing dot-product attention with a robust KDE approach to enable more robust transformer models. The experiments illustrate that the proposed approach can maintain performance on clean data while improving performance on contaminated data.

**Strengths:**

* Given the wide use of transformer models, the paper tackles an important topic, as robustness to training data contamination can improve their performance.
* The proposed approach is well motivated by the connection to non-parametric regression.
* The experiments show very promising results on both clean data and contaminated data, across
* The proposed approach does not introduce additional parameters.

**Weaknesses:**

Practicality of the approach for large datasets: the proposed approach incurs additional computational complexity even with the proposed approximations. The experiments are performed on small versions of the models. It's unclear how well this can be applied to large scale training of transformer models.

**Questions:**

N/A

**Limitations:**

The authors accurately describe limitations and potential solutions throughout the paper and in the Conclusion and Future Work section.

---

> ### Author Rebuttal · Authors · 2023-08-09
>
> Dear Reviewer `SN7k`,
>
>
> We would like to thank you for your valuable comments and positive feedback. We now address your major concern below.
>
> **Q: The proposed approach incurs additional computational complexity even with the proposed approximations. The experiments are performed on small versions of the models. It's unclear how well this can be applied to large-scale training of transformer models.**
>
> **A**: With respect to computational complexity, it's crucial to highlight that if we operate under the assumption of having a clean training set—where contaminated samples due to factors like distribution shifts, adversarial attacks, or data poisoning only manifest during the inference stage—then the robust attention mechanism only needs application during inference. This can lead to a significant reduction in the time spent calculating the robust attention vector during training. However, when we don't have prior knowledge about the quality of the data, contamination can potentially occur during both training and inference. In such cases, it becomes prudent to invest the extra computational time to shield against outliers throughout the entire pipeline.
>
> In terms of models in different scales, for the robust language modeling task outlined in Section 4.1, we've carried out assessments using both the small and medium versions of Transformers. You'll find these detailed in Table 5 in Appendix E, where the pertinent text is highlighted in red. Additionally, we've also contrasted the time complexity and memory usage of Vision Transformers employing tiny, small, and base Transformer backbones respectively - see Figure 4 and the accompanying discussion for this.
>
> We wish to emphasize that we've done our utmost to ensure that our experimental results are as thorough as possible. However, we were constrained by both time and the available computational resources at the time of submission. Before the rebuttal process begin, we were able to perform extra experiments using Transformers with greater capacities. The results of these investigations can be found in the general announcement.
>
> Please do not hesitate to let us know if you have additional questions.
>
> Sincerely,
>
> Authors

---

### Author Rebuttal · Authors · 2023-08-10

Dear Reviewers,

We greatly appreciate the time and effort you've dedicated to offering valuable feedback for the enhancement of our manuscript. We'd like to address a frequently asked question and provide details on our further investigations.

1. **Is this method purely post-hoc, or does it necessitate modifications during the training process? Put differently, is it feasible to train using the standard KDE and then employ a robust KDE only during testing?**

We aim to incorporate our robust attention mechanism into both the training and inference stages. Given the uncertainty about data cleanliness, there's a possibility of encountering contaminated samples at either stage. Thus, it is worthwhile to defend against outliers throughout the entire process.

In the training context, our methods modify the computation of attention vectors across each Transformer layer, making them less susceptible to contamination from outliers. Importantly, this entire process remains nonparametric, with no introduction of additional model parameters. However, the resulting attention vectors, whether shaped by re-weighting or the median-of-means principle, diverge from those generated by the standard softmax attention. This distinction influences the model parameters learned during training.

During inference, the test data undergoes a similar procedure to yield robust attention vectors. This ensures protection against potential outlier-induced disruptions within the test sequence.

However, if we assume the availability of a clean training set — where contamination arises solely from distribution shifts, adversarial attacks, or data poisoning during inference — it is sufficient to restrict the application of the robust attention mechanism to just the inference phase. This could considerably reduce the computational time required for robust attention vector calculation during training.

In our empirical evaluation, we engaged the robust attention mechanism throughout both phases and recorded the associated computational time during training. We found that on standard datasets like ImageNet-1K, WikiText-103, and the UEA time-series classification, infusing the robust attention led to a drop in training loss. This suggests that training data itself may contain inherent noise or outliers.

2. **Large-scale experiments and computational time.**

We wish to emphasize that we've done our utmost to ensure that our experimental results are as thorough as possible. However, we were constrained by both time and the available computational resources at the time of submission. Before the rebuttal process begin, we were able to perform extra experiments using Transformers with greater capacities.

We have also monitored the computational time for each task in the paper. As mentioned in the former question, the computational time can be greatly reduced if the robust attention mechanism is only applied in the inference stage.

Please find the additional results in the attached PDF file.

Sincerely,

Authors

---

### Comment · Area_Chair_MJUp · 2023-08-13
**Please respond to author rebuttals**

Dear Reviewers,

Please respond to authors after carefully reading the author rebuttals and other reviews. If your assessment of the paper changes, please update your score with a short justification for the new rating. You are welcome to initiate discussions with authors or other reviewers.

Thank you,
AC

---

### Decision · Program_Chairs · 2023-09-21

**Decision:**

Accept (poster)

**Comment:**

The paper proposes to use robust kernel density estimation (KDE) in place of the standard dot-product self-attention in Transformers to improve the robustness of models on corrupted data in both vision and language tasks. All five reviewers rated this paper as above acceptance threshold or more positive. Reviewers mentioned that the re-interpretation of self-attention as KDE is a novel perspective, the motivation of using robustness KDE is strong and the empirical evidence of improved robustness is extensive. The weaknesses brought by the reviewers are partly addressed in rebuttal and outweighed by the merits. The AC is pleased to recommend acceptance of this paper.